# Studies on geochemical characteristics and biomineralization of Cambrian phosphorites, Zhijin, Guizhou Province, China

**Lei Gao[1,2], Ruidong Yang[1,2]\*, Tong Wu[3], Chaokun Luo[1,2], Hai Xu[4], Xinran Ni[1,2]**

**1** College of Resources and Environmental Engineering, Guizhou University, Guiyang, Guizhou, China,
**2** Key Laboratory of Karst Georesources and Environment, Ministry of Education, Guizhou University,
Guiyang, Guizhou, China, **3** Resource and Environmental College, Henan University of Engineering,
Zhengzhou, Henan, China, **4** Department of Tourism and Geography, Tongren University, Tongren,
Guizhou, China

\* rdyang@gzu.edu.cn

**Data Availability Statement:** All relevant data are within the paper and its Supporting information files.

## Abstract

Phosphate rocks, an important ore resource in Guizhou Province, China, are mainly hosted within the Sinian Doushantuo Formation and the Cambrian Meishucun Formation. In addition, the phosphate rocks of the Cambrian Meishucun Formation are rich in biological fossils. Although numerous studies investigating the genesis of phosphate deposits have been performed, the relationship between biological activity and the formation of phosphate deposits in the lower Cambrian Meishucun Formation has not been convincingly explained. This study focuses on the biological fossil assemblage, the characteristics of phosphorus, and the relationship between biological and phosphorus enrichment of the lower Cambrian phosphorites. The primary objectives of our study are to analyze the role of organisms in the formation of phosphorites, restore the phosphorus-formation environment of the Cambrian Meishucun Formation, and construct a sedimentary model of the phosphorites in the Meishucun Formation. The results indicate that there is a significant positive correlation between biological activity and the deposition of phosphorites, that is, the higher the degree of biological enrichment and differentiation, the stronger the deposition. The geochemical analysis of several profiles in the Zhijin phosphorite block shows that the phosphorite block was deposited in an oxygen-rich environment and was affected by a high-temperature hydrothermal fluid. Upwelling ocean currents supplied abundant phosphorus and other nutrients, which provided the conditions for small shells and algae to flourish. Biochemical activity was a crucial factor in the deposition of the phosphorite.

## Introduction

As an important ore resource in Guizhou Province, China, the phosphate rocks mainly occur in the Sinian Doushantuo Formation and the Cambrian Meishucun Formation. In addition, the phosphate rocks of the Cambrian Meishucun Formation commonly contain rich biological fossils, which is of great significance for the study of Cambrian fossils and to stratigraphic

**Funding:** This research was supported by the National Natural Science Foundation of China (41890841, U1812402), the Project of The Department of Science and Technology of Guizhou Province (Guizhou Science and Technology Cooperation Platform Talents [2018]5613), the Study on metallogenic regularity and prospecting prediction of rare earth, barium, fluorine and other special resources in Guizhou ([2022]ZD004), and the Postgraduate Innovation Fund of Guizhou Province (Guizhou Education Cooperation YJSCXJH [2019]040). The funders had no role in study design, data collection and analysis, decision to publish, or preparation of the manuscript.

**Competing interests:** The authors have declared that no competing interests exist.

division and comparison. The major episodes of phosphorite accumulation were related to the variations caused by the biogeochemical cycling of P through time, which in turn were connected to changes in the paleogeography, paleoceanography, and biological evolution [1]. Most previous studies on the relationship between phosphorites and organisms have focused on traditional paleontology and the burial of phosphorites [2–7]. It is believed that one method by which organisms form phosphorus is the direct accumulation of phosphorus shells [8]. Algal organisms also play a key role in the accumulation of phosphorus in shallow sea areas [9, 10]. The variable paleo-ocean environment and the enrichment of phosphorus, oxygen, and sunlight promote the flourishing of a large number of organisms and the occurrence of large-scale global phosphorus formation events [11]. The organisms in shallow seas affect the formation of phosphorus blocks through adsorption and desorption of phosphorus. However, some scholars believe that the organisms themselves contribute little to the accumulation of phosphorus, which is mainly related to carbonate-fluorapatite and collophanite formed by chemical precipitation in phosphoric water bodies as a result of rising ocean currents [7].

Geochemical methods are some of the most important methods in geological research. Previous studies [e.g., 12–14] have shown that the phosphorites in the lower Cambrian strata have a marine sedimentary origin. However, based on trace elements and fluid inclusions, some studies have suggested that the characteristics of the phosphorites in the lower Cambrian strata are typical of hydrothermal deposition [15–17]. Hydro-thermal activity can transform protogenetic minerals [16–19] and affect the formation of REE-rich phosphorous deposits. However, the relationship between biological activity and the formation of phosphate deposits in the lower Cambrian strata in China remains unclear. With this in mind, the geochemical characteristics of the Cambrian phosphorites in Zhijin (Guizhou, China), the characteristics of the biological assemblages in the different sedimentary microfacies, and the role of organisms in the phosphorization process were investigated in this study in order to analyze the role of organisms in the formation of phosphorites, enrich the theory of biological mineralization, restore the phosphorus-formation environment of the Cambrian Meishucun Formation, and construct a sedimentary model of the deposition of the phosphorites in the Meishucun Formation.

## Geological setting

The Zhijin phosphate deposit region was formed in the Meishucun period of the Cambrian [20]. This deposit contains large-scale, high-quality phosphorus block rocks and represents the most important phosphorus formation period after the deposition of the phosphorus block rocks in China during the Doushantuo period [20–23]. The Zhijin phosphate ore area is located in the western-central part of Guizhou Province and at the southwestern end of the Yangtze Platform [22, 24]. The sampling sites for this study are located in six mining areas in Zhijin (Xiongjaichang (XJC), Lianxing (LX), Yinchanggou (YCG), Damachang (DMC), Gaoshan (GS), and Gezhongwu (GZW)), and the research areas are mainly distributed in the northwestern limb of the Guohua and Zhangwei anticlines (Fig 1) [3]. The faults are simple, predominantly normal faults, and the small local structures are well developed. Faults $F_1$ (normal fault), $F_2$ (reverse fault), and $F_3$ (reverse fault) control the strike of the entire Zhijin phosphate ore area and have damaged the integrity of some of the ore beds, affecting the output form of the phosphate ore in the study area. The exposed strata mainly include the Sinian Dengying Formation and the Cambrian Gezhongwu and Niutitang formations. The phospho-bearing rock assemblages of siliceous blocks and phospho-blocks, the carbonates, and the phospho-block constitute the largest rare-earth phosphate deposit formed in the Meishucun period of the Cambrian in Guizhou Province, China. The different sampling areas in the

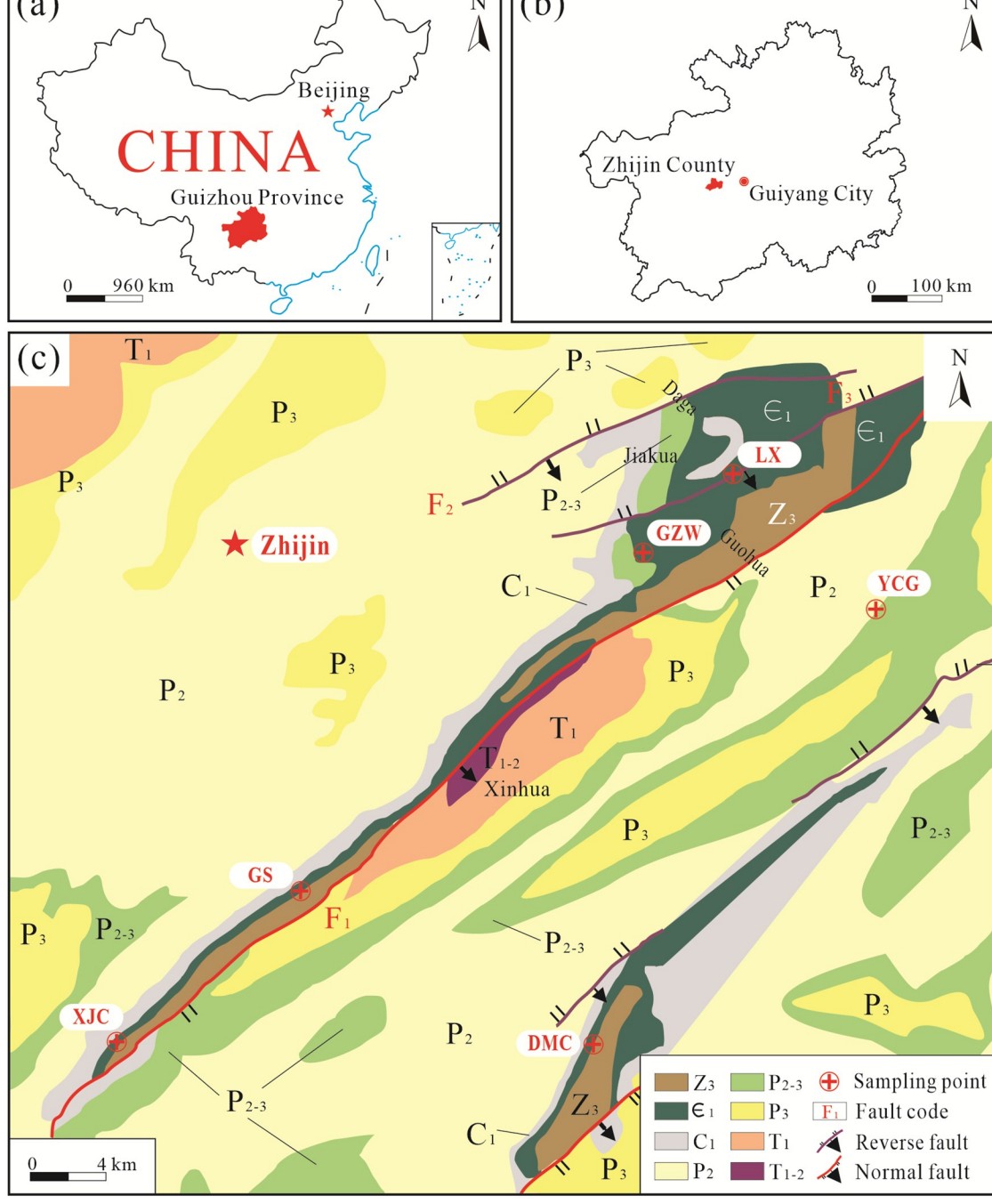

**Fig 1.** Map of China (a), Map of Guizhou Province (b), Regional geologic map of the Zhijin area showing the six sampling profiles, including the Xiongjaichang (XJC), Lianxing (LX), Yinchanggou (YCG), Damachang (DMC), Gaoshan (GS), and Gezhongwu (GZW) locations (c). (a) and (b) were created using free vector map data from the Public Domain (CC0), derived from USGS National Map Viewer (http://viewer.nationalmap.gov/viewer/). (c) modified after Fig 1 in ref. [3].

Zhijin research area exhibit different sedimentary characteristics, and the different sedimentary environments also provided different living environments for the growth of organisms.

The lower Cambrian phosphate deposits in southern China are the products of transgression and upwelling of ocean currents. Previous studies have shown that during this period, the

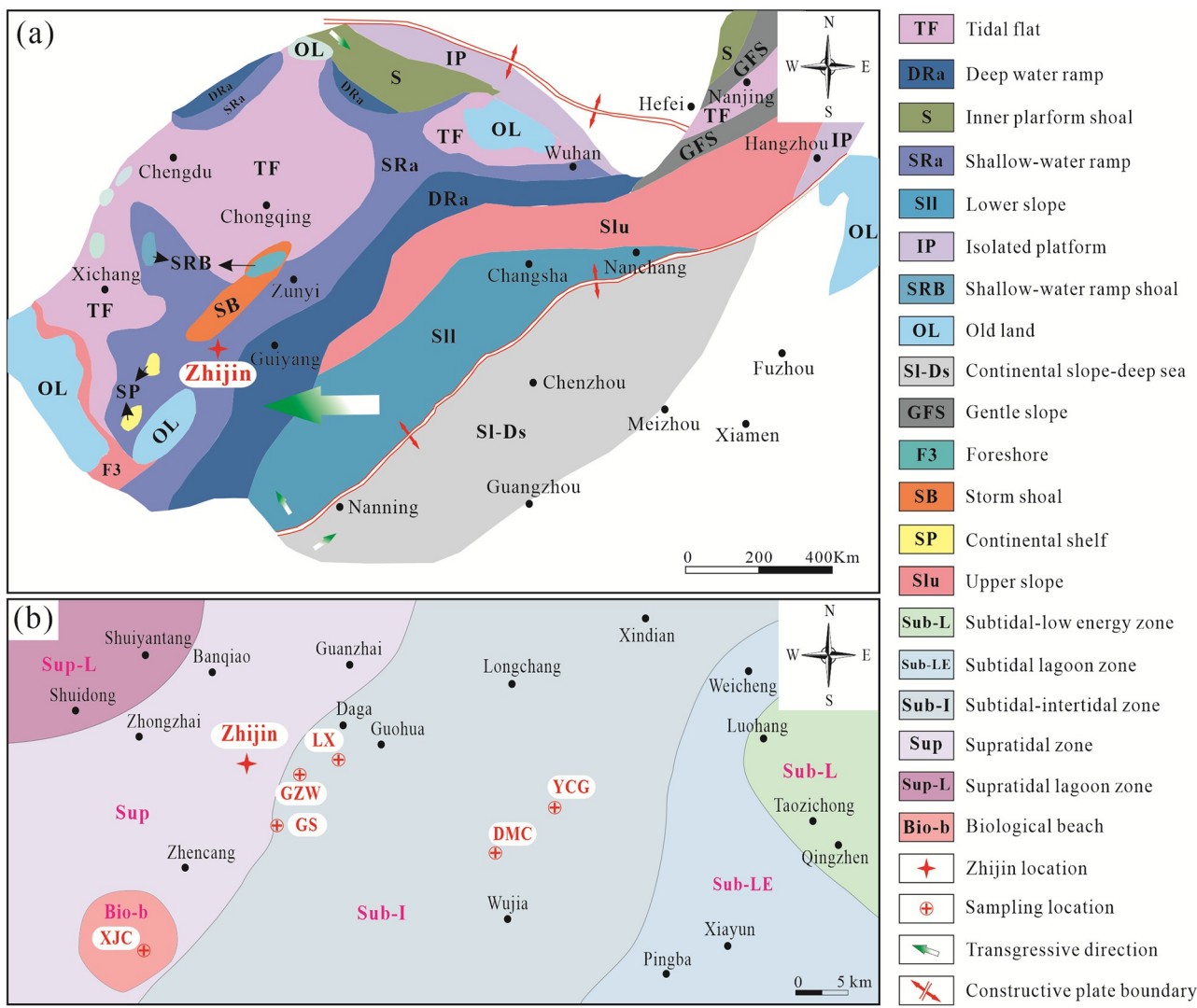

**Fig 2.** Lithofacies and paleogeography of the lower Cambrian strata in Zhijin showing: (a) Lithofacies and paleogeography of the lower Cambrian strata in South China Block; (b) Map of the paleogeographic zoning of the lower Cambrian phosphorite rocks in Zhijin area, including the Xiongjaichang (XJC), Lianxing (LX), Yinchanggou (YCG), Damachang (DMC), Gaoshan (GS), and Gezhongwu (GZW) sampling locations. (a) modified after Fig 1 in ref. [21]. (b) modified after Fig 3–9 in ref. [24].

sedimentary facies included a littoral tidal flat–shallow water slope (phosphate-bearing carbonate slope), a deep-water slope (phosphatic concretion-bearing siliceous slope), and a continental slope deep water basin from southwest to east (Fig 2a) [20–22]. During the Meishucun period of the lower Cambrian in Zhijin, the depth of the sea water increased continuously during a transgression event [20, 22]. A large amount of phosphatic material was supplied by the stronger currents in the Meishucun period [20, 21, 24]. The deposition process was controlled by many factors, including the biological chemistry and sedimentary environment [20–23]. The Zhijin area formed a large-scale phosphorus rock series, which was controlled by the sedimentary facies and can be divided into an intertidal low energy phosphorus forming zone, a subtidal-intertidal high energy phosphorus forming zone, a supratidal shoal phosphorus forming zone, and a water body that gradually became shallower (Fig 2b) [24].

A columnar comparison chart of the lower Cambrian phosphorus-bearing rock systems in Zhijin is shown in Fig 3a. The DMC and YCG areas of Zhijin the are located in the low-energy sedimentary environment of the subtidal-intertidal zone. The phosphorus blocks are mainly dolomitic and siliceous phosphorus blocks (Fig 3b and 3c), which contain lens-like structures

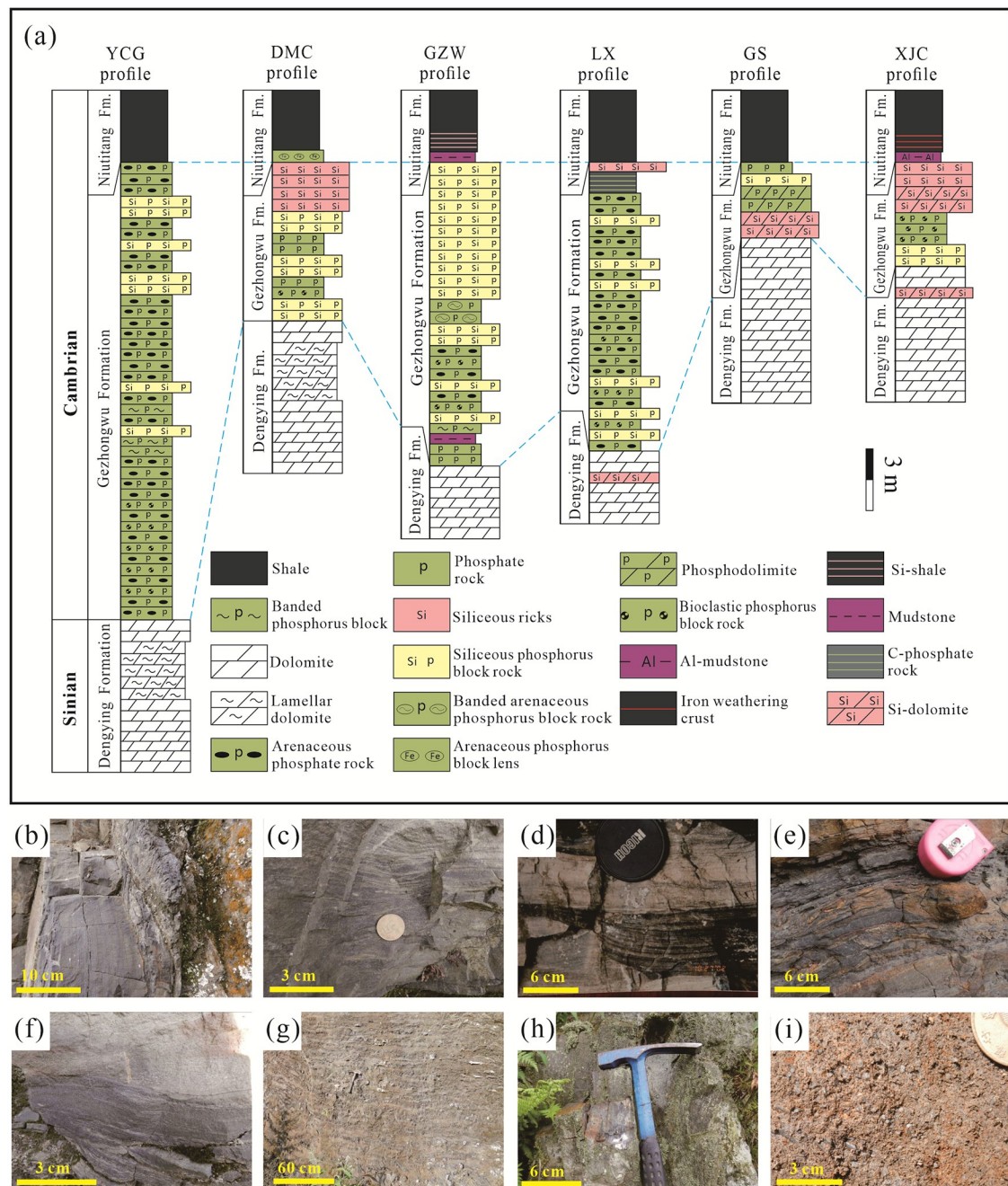

**Fig 3.** Lithostratigraphic columns and field photographs of the lower Cambrian Phosphate rocks in Zhijin area showing: (a) Columnar comparison of the lower Cambrian phosphorus-bearing rock systems in Zhijin area; (b) Phosphorus-bearing argillaceous dolomite; (c) Lamellar siliceous phosphorite; (d) Lenticular bedding; (e) Thin clastic phosphorite; (f) Clastic phosphorite with low-angle cross-bedding; (g) Interbedding phenomenon of phosphorite; (h) Siliceous phosphorite with horizontal texture; (i) Biophosphorite.

(Fig 3d), and were formed where the hydrodynamic conditions were relatively weak. The LX and GZW areas of Zhijin are located in the high-energy sedimentary environment of the subtidal-intertidal zone, which are dominated by clastic phosphorus blocks and bioclastic rocks (Fig 3e and 3f). Compared with the siliceous phosphorus block, due to the lower weathering strength of the clastic phosphorus block and the repeated erosion caused by tidal action, regular and obvious interbeds (Fig 3g) formed in this region, which has the best phosphorus formation conditions and the formation of large phosphorus block deposits. The GS area of Zhijin is located in the transition zone between the intertidal zone and the supratidal zone, and it is characterized by low hydrodynamics and a siliceous phosphorus block with a horizontal texture (Fig 3h). The XJC area of Zhijin is located in the supratidal zone and contains a biological shoal. The phosphorus blocks mainly include biological phosphorus blocks and bioclastic phosphorus blocks (Fig 3i).

## Material and methods

### Permission statement

No permits were required for the described study, which complied with all relevant regulations.

### Fossil disposal

After soaking the phosphorus rock specimens with glacial acetic acid, some of the small shell fossils (SSFs) exfoliated from the phosphorus rock. The exfoliated individual fossils were identified under a microscope, fixed in place using Canadian gum, sliced, and processed to make thin sections. For the SSFs that could not be exfoliated from the phosphorus block rock, the entire section of the phosphorus block rock was made into thin sections to observe the cross section, vertical section, and oblique section of various fossils. These thin sections were placed under a polarized light microscope for comparative study using single polarized light and orthogonal polarized light to better observe the mineral compositions and the microscopic structures of the SSFs. The micromorphological observations and microchemical analysis of the SSFs were performed using a HITACHI-SU8010 scanning electron microscope (SEM) equipped with an energy dispersive spectrometer (EDS). Before the testing of SEM-EDS, the fossil shells and thin sections were sprayed with a thin layer of gold. The mineralogical composition of the SSFs, bioclastic phosphate rocks, siliceous phosphorite blocks and carbonaceous phosphorite blocks were performed using an Empyrean powder X-ray diffractometer (XRD, PANalytical B. V.) with the following operating conditions: Cu Kα radiation at a voltage of 45 kV and a current of 40 mA, scanning speed of 8°/min, and 2-theta (°) ranges from 5°~70°. The optical microscopy, SEM-EDS and XRD were carried out at Guizhou University, People's Republic of China.

### Geochemical analysis

The major elements of 12 samples were analyzed at the ALS Mineral Lab (Guangzhou). The powdered samples (~200 mesh) were mixed with $Li_2B_4O_7\text{-}LiBO_2$ and was fully melted in a 1000°C furnace. After the melt cooled, dilute $HNO_3$ and dilute HCl were added to dissolve the melt, and then, the quantitative analysis was conducted via X-ray fluorescence staining using a PANanalytical AXIOS instrument. The spectral interference between the elements was corrected to obtain the final analysis results, with an analysis error of better than 2%. The trace and rare earth elements of 41 samples were analyzed by using Quadrupole inductively coupled plasma-mass spectrometer (Q-ICP-MS, ELAN DRC-e, PerkinElmer, Canada). The analysis

method was follows: the powdered samples (~200 mesh) were digested with HCl, $HNO_3$, HF and $HClO_4$, after dryness, prepared samples were dissolved with HCl, and then analyzed by Q-ICP-MS. The detection limits of trace and rare earth elements were: for V, Ni, Ag, As and Sb 0.01 ppm; For Y and REE 0.01 to 0.05 ppm. The trace and rare earth elements of samples were analyzed at the Institute of Geochemistry, Chinese Academy of Sciences. The results of whole-rock major, trace and rare earth elements concentrations in all samples are reported in S1–S3 Tables. The REE parameters analyzed are as follows: $\delta Ce = Ce/Ce^* = Ce_N/0.5(La_N+Pr_N)$ [25], $\delta Eu = Eu/Eu^* = Eu_N/(Sm_N \times Gd_N)^{0.5}$ [26], $Pr/Pr^* = Pr_N/0.5(Ce_N+Nd_N)$ [25], and $Y/Y^* = 2Y_N/(Dy_N+Ho_N)$ [27], where N denotes normalization relative to Post-Archean Australian Shale (PAAS) [26].

## Results

### Biological assemblage characteristics

By slicing the Meishucun phosphorus block rock from Zhijin, the species of small shell animals in the phosphorus block rock were identified. The sedimentary environments of the DMC and YCG areas were in the subtidal-intertidal zone, low-energy environment. The rock series in these two areas is rich in phosphorus and contains a large number of siliceous sponge spicules (Fig 4a), while the other types of SSFs are scarce. In these profiles, the phosphorus siliceous mudstone was found to contain some large Hyolithes fossils (Fig 4b) and a small number of Zhijinitid fossils (Fig 4c and 4d). In general, the hydrodynamic environment in which sponges live is relatively weak. The occurrence of a large number of sponges in this area also indicates that the dark gray siliceous phosphoric rocks in the YCG and DMC areas are formed in a subtidal environment with weak hydrodynamic forces. Combined with the biological fossil assemblage and the characteristics of the phosphorite, the low tidal energy environment was not conducive to the deposition of phosphorite because the biota was not prosperous and the biological and biochemical interactions were weak, which was not conducive to phosphorus precipitation.

The sedimentary environments of the LX and GZW areas were the high-energy subtidal-intertidal zone, which was characterized by strong hydrodynamic forces and frequent tidal action. A large number of phosphorous deposits were formed in this area. Because the water was rich in phosphorus and other nutrients, the shallow water, moderate temperature, and sunny conditions made the small shell fauna flourish in this area, leading to biological abundance and a high degree of differentiation. In particular, the abundances of Hyolithes (Fig 4e), Zhijinitids (Fig 4f), Hyolithelminthida (Fig 4g), Parazhijinites (Fig 4h), and spherical shells (Fig 4i) are extremely high. However, the algae and sponge contents are lower in low-energy environments. The analysis of the biological abundance, diversity, and preservation of the fossils suggests that there was a close relationship between phosphate deposition and biota development in this area. The shallow sea environment where the biota is abundant has biological and biochemical effects, which are conducive to phosphate deposition and the formation of a large number of phosphate deposits.

The sedimentary environment of the GS area was the low-energy transitional environment between the intertidal and supratidal zones, which was characterized by weak hydrodynamics and poorly developed phosphorite-bearing rocks. The phosphorite-block rocks mainly contain a large number of macroalgal (Fig 4j–4l), sponge spicule (Fig 4m), and Hyolithes fossils (Fig 4n). The environment where the sponges and algae lived had relatively weak hydrodynamics, while the content of Hyolithes fossils, which flourish under strong hydrodynamic conditions, was low. Combined with the sedimentary characteristics, this region was a low-energy

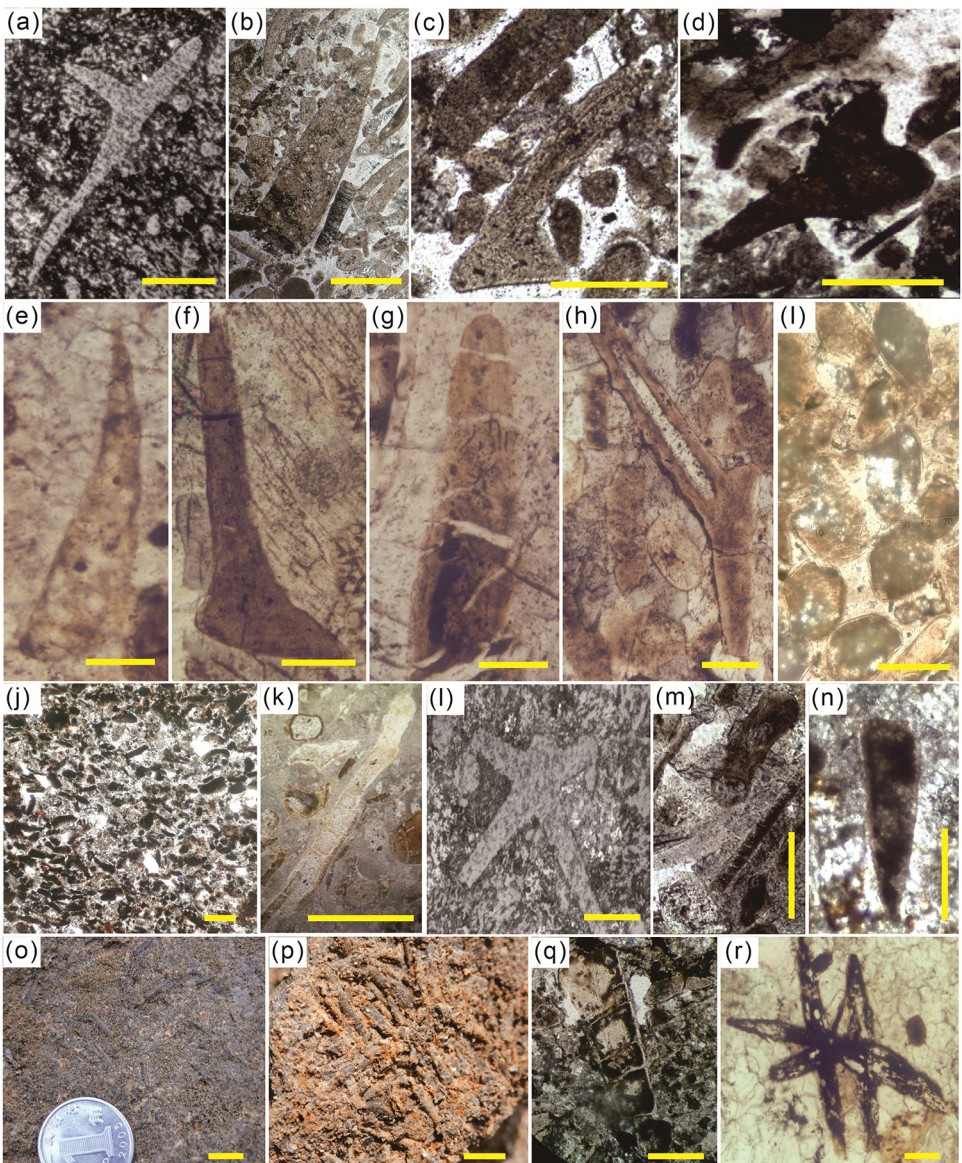

**Fig 4.** Fossil assemblages in the Zhijin phosphorites showing: (a) Sponge spicule fossils; (b) Hyolithes fossils; (c, d) Zhijinitid fossils; (e) Hyolithes fossils; (f) Zhijinitid fossils; (g) Hyolithelminthid fossils; (h) Parazhijinite fossils; (i) Spherical fossils; (j) Abundant macroalgal and sponge spicule fossils; (k) Macroalgal fossils; (l) Sponge spicule fossils; (m) Macroalgal and sponge spicule fossils; (n) Hyolithes fossils; (o, p) Abundant Hyolithes fossils in the phosphorite; (q) Cross-section of Hyolithes fossils; (r) Sponge spicule fossils. The scale bars are 200 μm for a-d, 100 μm for e-i, 200 μm for g-n, 1 cm for o-p, and 150 μm for p-r.

sedimentary environment with weak hydrodynamic conditions, phosphorite was not developed, and the rock strata are thin and dark in color.

The sedimentary environment of the XJC area was the supratidal zone, which was characterized by water with a relatively high phosphorus content and salinity, moderate oxygen and sun light, frequent tides, and stronger water power. This was favorable to the development of small shell animals, providing comfortable conditions for breeding and rich material sources. The abundances of the swimming and floating small shell animals were uniform and were

dominated by Hyolithes fossils, accounting for about 70%, with larger individual fossils (up to 2 cm) (Fig 4o–4q), sponge spicule fossils (Fig 4r), and some uncertain kinds of fossils. Zhijinitid fossils were not observed. The Hyolith fossil shells are thick and well preserved, which indicates strong wave resistance and benthic organisms.

## Elemental compositions of SSFs

The SEM-EDS analysis of the SSFs in the Zhijin area revealed that although the SSFs have different microstructures, the main mineral compositions of the shells are consistent. The results indicate that the fossil shells are mainly composed of C, O, P, F, and Ca (Fig 5a–5d). The C, O, F, P, and Ca contents are about 8.98–18.09 wt%, 25.78–50.15 wt%, 3.65–6.93 wt%, 9.59–16.70 wt%, and 17.82–39.28 wt%, respectively, while the contents of the other elements, including Al, Si, Fe, and S, are low. These elements are not shown in the energy spectrum. Meanwhile, the compositions of the shells of the SSFs in Zhijin are uniform, without the secondary enlargement caused by late hydrothermal alteration. These evidences indicate that the primary mineral composition of the SSFs is carbonate-fluorapatite. Second, it was found that some of the collophanite cements do not contain phosphorus (Fig 5d), and all of the phosphorus exists in the shells of the SSFs, forming biophosphorus block deposits.

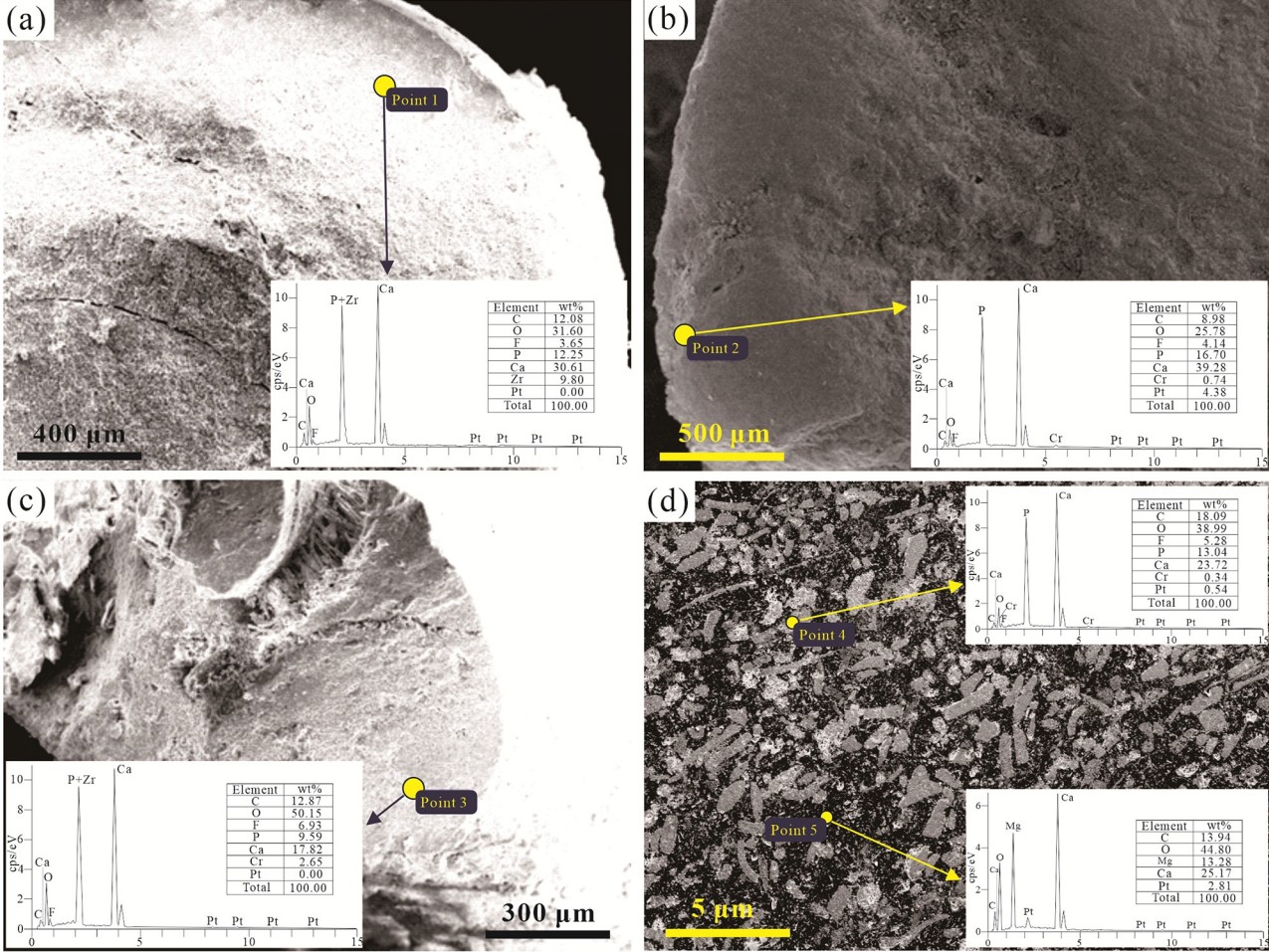

**Fig 5.** SEM-EDS analysis of the fossil shells of the SSFs in Zhijin area showing: (a-c) Hyolithes fossils; (d) A thin section of Hyolithes fossils.

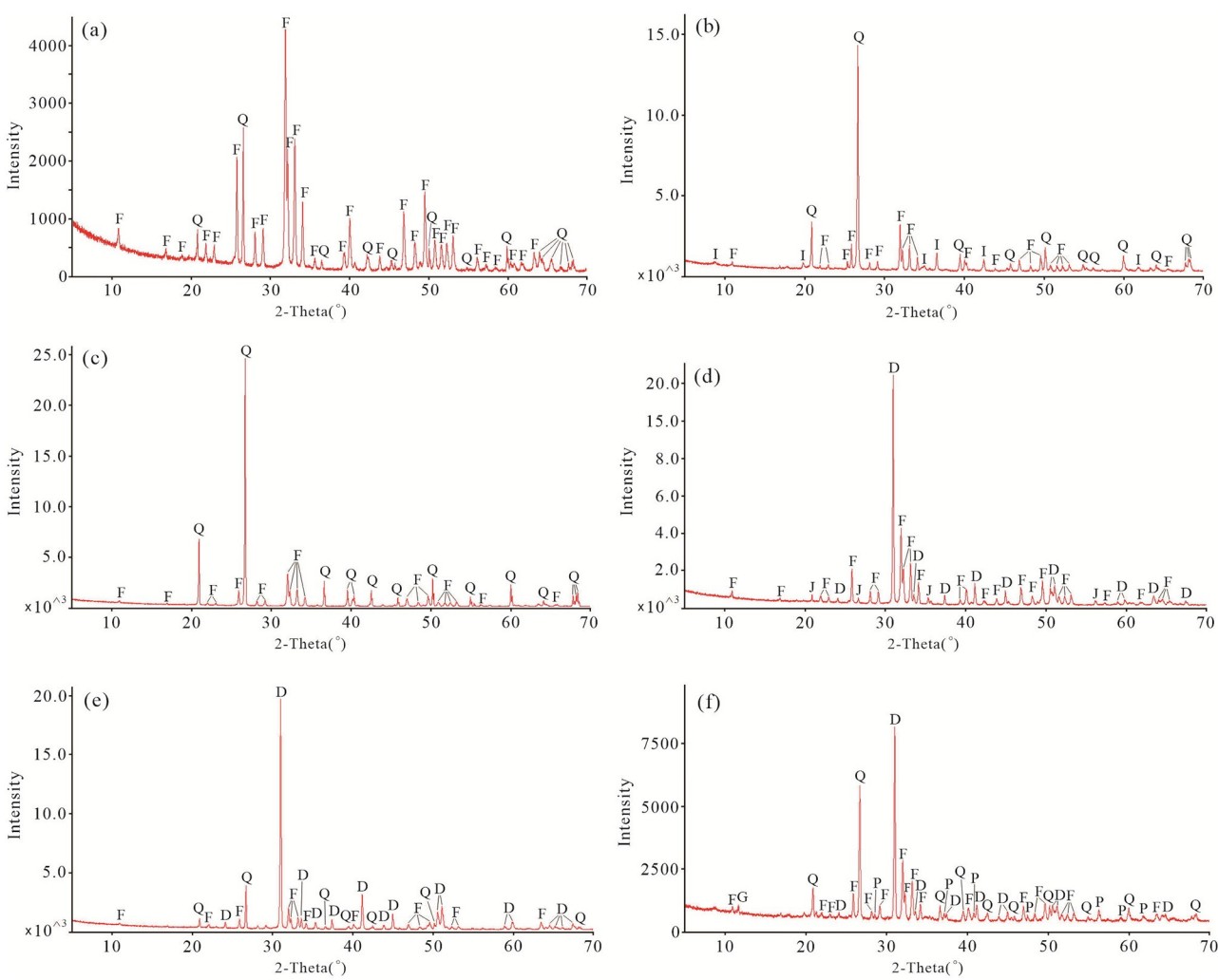

**Fig 6.** Powder-XRD patterns of the representative small shell fossils (SSFs), bioclastic phosphate rocks, siliceous phosphorite blocks, and carbonaceous phosphorite blocks in the Zhijin area: (a) SSFs of the XJC profile; (b) Bioclastic phosphate rocks of the DMC profile; (c) Bioclastic phosphate rocks of the LX profile; (d) Bioclastic phosphate rocks of the YCG profile; (e) Siliceous phosphorite blocks of the GZW profile; (f) Carbonaceous phosphorite blocks of the LX profile. F = Carbonate-fluorapatite, Q = Quartz, I = illite, D = dolomite, J = jeremeievite, P = pyrite, G = gypsum.

## Mineralogical characteristics

The results of XRD analysis indicate that the SSFs are mostly composed of carbonate-fluorapatite, and small amount of quartz (Fig 6a). The bioclastic phosphate rocks in the DMC profile, LX profile and YCG profile are mainly composed of quartz and carbonate-fluorapatite, followed by illite, dolomite and jeremeievite (Fig 6b–6d). In contrast, the siliceous phosphorite blocks of the GZW profile consist of dolomite, quartz and carbonate-fluorapatite (Fig 6e), and the carbonaceous phosphorite blocks of the LX profile consist of dolomite, quartz, carbonate-fluorapatite, pyrite and gypsum (Fig 6f). According to the XRD analysis, and combined with the SEM-EDS analysis, the mineralogical composition of the SSFs is dominated by carbonate-fluorapatite.

## Geochemical characteristics

Based on the extensive geochemical analysis of samples of phosphorite block rocks from Zhijin (S1 Table), it was found that the phosphorus content did not change significantly, and only

some samples from the DMC and YCG areas exhibited a sudden decrease in the phosphorus content. The LX, GS, and GZW areas are characterized by high $SiO_2$, and the $P_2O_5$ contents of the LX phosphorites range from 16.26% to 22.43%. Combined with the lithologic characteristics of the profiles (Fig 3), it is concluded that the phosphorus content of the siliceous phosphorites is slightly higher than that of the bioclastic phosphorites and arenaceous phosphorites, which have a higher biomass. The $P_2O_5$ content of the GS phosphorites is 19.56%, and the phosphorite-bearing rock series is mainly siliceous phosphorite blocks (Fig 3); thus, the phosphorus content should remain basically unchanged. The $P_2O_5$ content of the GZW profile phosphorites varies greatly, ranging from 13.90% to 35.69%. The $P_2O_5$ content of the upper dense block phosphorites is significantly higher than those of the bottom siliceous phosphorites, which contain a large number of SSFs. The $P_2O_5$ content of the DMC section is very low, less than 5%. The $P_2O_5$ content of the YCG profile varies greatly. The $P_2O_5$ content of the bottom of the profile is greater than 25%, while the $P_2O_5$ content of the top of the profile is less than 10%.

Based on the above mentioned results, by means of rare earth and trace element geochemistry, the sedimentary environment of each sampling profile in the Zhijin study area was further traced. The analysis results are presented in S2 and S3 Tables. The REE concentrations of the XJC strata vary greatly and are mainly concentrated in the phosphorite rocks. The highest total REE content ($\Sigma$REE) is 488.53 ppm, the lowest is 1.82 ppm, and the average is 212.09 ppm (S3 Table). The degree of REE enrichment of the GS strata is generally lower than those of the other areas, with $\Sigma$REE values ranging from 63.34 ppm to 532.19 ppm and an average value of 201.94 ppm (S3 Table). The $\Sigma$REE values of the LX strata range from 336.92 to 730.12 ppm, with an average of 561.61 ppm (S3 Table). The $\Sigma$REE values of the YCG strata range from 294.44 to 784.65 ppm, with an average value of 605.25 ppm, indicating a high degree of enrichment (S3 Table). The degree of REE enrichment of the GZW strata is high, with $\Sigma$REE values ranging from 209.83 to 794.86 ppm and an average of 554.74 ppm (S3 Table). The degree of REE enrichment of the DMC strata is slightly lower than those of the other areas, with $\Sigma$REE values ranging from 18.24 to 598.86 ppm and an average value of 256.09 ppm (S3 Table). In addition, the MREE are enriched relative to the LREE and HREE in the Zhijin phosphorites samples. The average $(MREE/LREE)_N$ value is 2.34 and the median value is 2.26. The average $(MREE/HREE)_N$ value is 1.38 and the median value is 1.39 (S3 Table).

In addition, the values of some of the geochemical parameters for the Zhijin phosphorites samples were calculated and are presented in S3 Table. The $Eu/Eu^*$ and $Ce/Ce^*$ values are 0.92–3.83 and 0.19–0.91, respectively (S3 Table). The $Pr/Pr^*$ and $Y/Y^*$ values are 1.10–1.62 and 0.44–1.77, respectively (S3 Table). The La/Nd, Y/Ho, Er/Nd, and V/Ni ratios of the phosphatic rocks are 0.36–1.65, 12.07–46.53, 0.09–0.15, and 0.06–44.4, respectively (S3 Table).

## Discussion

### Relationship between biological assemblage characteristics and sedimentary facies belt

**SSFs abundance and diversity statistics.** The paleogeography of the lower Cambrian strata in the Zhijin area shows that from the DMC and YCG areas to the XJC area, the low-energy subtidal-intertidal zone transitioned to the high-energy supratidal zone and the water depth became shallower, forming biological shoals as far as the XJC area [21, 24]. The LX and GZW areas were the sedimentary center and were the most beneficial areas for deposition of metallogenic phosphorus deposits [24]. Zhu et al. [4] and Qian et al. [2] concluded that most of the SSFs in the lower Cambrian strata experienced heterochthonous burial, but their transport action was weak. The SSFs preserved in the Zhijin phosphate rock were affected by high-

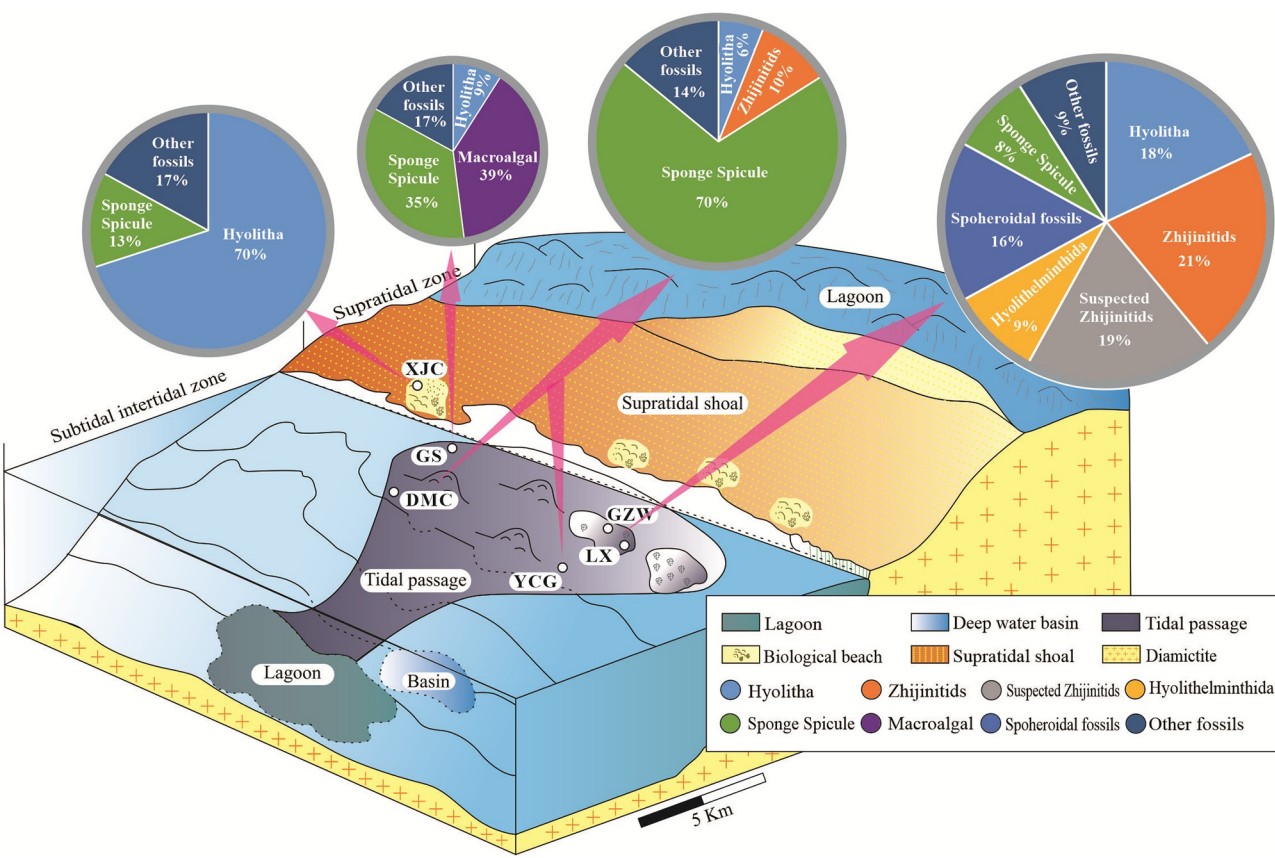

**Fig 7. Relationships between the biological assemblage characteristics and the sedimentary environment for the Zhijin phosphate deposit.**

energy hydrodynamics, and the integrity of the fossil preservation is poor, which also indicate heterochthonous burial. In order to investigate the effects of the different sedimentary facies belt on the biological development, we calculated the abundances and diversity of the SSFs in six study areas in the Zhijin phosphate ore region (Fig 7).

**Special development of SSFs.** There are obvious differences in the species and proportions of Hyolithes fossils in the different sedimentary environments. In the low-energy environments such as the DMC, YCG, and GS areas, the proportion of Hyolithes is significantly smaller. In terms of type, they are all Circothecimorpha class, Cirothecida order Hyolithes without lip development (Fig 8a–8e). The shells are small, the mouth of the shells is gentle, they are mostly round and oval in cross-section, and there is no distinction between the dorsal and abdomen. However, in the high-energy environments, such as the XJC, LX, and GZW areas, the Hyolithes fossils are relatively abundant, accounting for a high proportion of the fossils. In particular, in the biological shoal zone in the XJC area, the number and proportion of Hyolithes is the highest, and there are Hyolithimorph fossils with lip development (Fig 8f–8i). These results indicate that the Hyolithes adapted to the complex and changeable sedimentary environment and had a strong environmental adaptability. In addition, there are also Turcutheca fossils with multiple shell layers in the XJC area (Fig 8j–8l), with a "funnel in funnel" structure [28]. The shells are round-tubular, and there is a multi-layer structure similar to a casing inside, which is similar to fossils of the Ediacaran Cloudina Hartmannae [29]. In addition, Paracircotheca fossils with a gas chamber structure were also observed in the XJC area

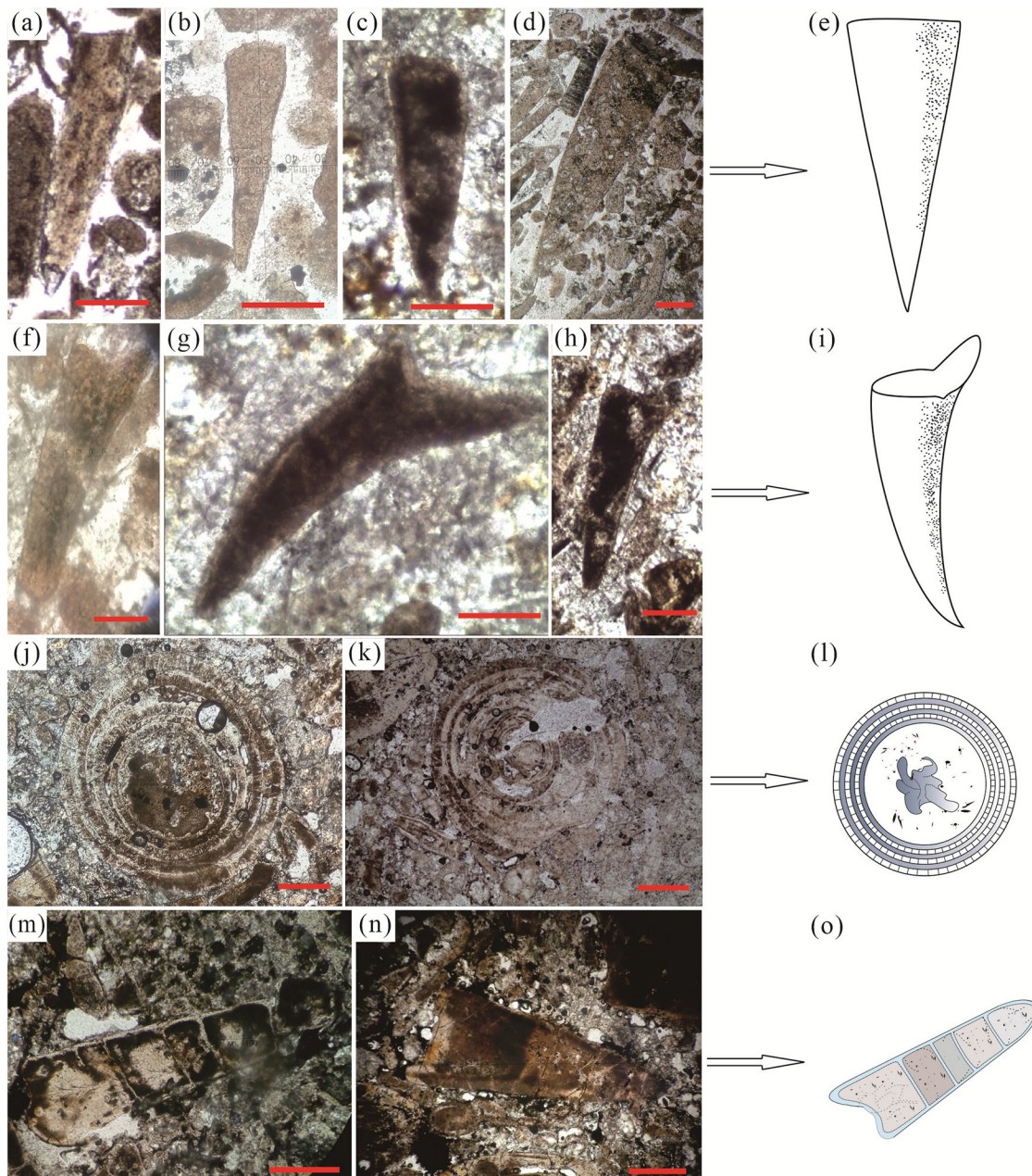

**Fig 8.** Special developmental phenomena of the SSFs in the phosphorite in Zhijin area showing: (a-d) Cirothecid fossils; (e) Pattern of the Cirothecid fossils; (f-h) Hyolithimorph fossils; (i) Pattern of the Hyolithimorph fossils; (j-k) Turcutheca fossils; (i) Pattern of the Turcutheca fossils; (m-n) Paracircotheca fossils; (o) Pattern of Paracircotheca fossils. The scale bars are 100 μm for a-o.

(Fig 8m–8o). It is related to the invertebrates Nautilus [30] and Ammonite [31], which have very similar gas chamber structures. The special environmental characteristics of the XJC area and the special development and growth of small shell animals indicate that the number of SSF shell unit layers and gas chamber structures are closely related to the biological growth environment. The small shell animals preferred the high-energy environments with strong hydrodynamics for breeding [4]. Salinity, nutrients, sunlight, and oxygen were the main factors restricting the breeding of a large number of species [4, 6].

### Sedimentary environment of phosphorites

**Ce anomaly.** Phosphorus blocks are mainly formed in two types of sedimentary basins, epicontinental sea and marginal sea basins. Phosphorus blocks formed in an epicontinental sea have no or positive Ce anomalies, while those formed in a marginal sea mostly exhibit negative Ce anomalies [32]. Meanwhile, the distribution of REY in the phosphorite is mainly controlled by the sedimentary environment, but irrespective of grain size in the rock [33]. By comparing the profiles of the Zhijin samples, it was found that the negative Ce anomalies (δCe) of the phosphatic samples (S3 Table) range from 0.23 to 0.91 (mean of 0.48). It was concluded that the lower Cambrian phosphorite should have formed in a marginal sea sedimentary environment. Ce anomalies can be used as an effective tracer for determining the redox conditions of the depositional environments of sediments and sedimentary rocks [34, 35]. Since the values of Ce anomalies can be overestimated due to a high concentration of La [25] or to the artificial calculations [36], it is necessary to discuss the real Ce anomaly values. Since no chemical factor will result in Nd or Pr anomalies, the existence of real Ce anomalies will result in Pr/Pr* values that are not less than 1 [25]. Therefore, Pr/Pr* vs Ce/Ce* bivariate plots [23, 25, 37] were used to assess the degree to which La effected the Ce anomaly values of the Zhijin phosphorites (Fig 9a). It was found that the Ce anomalies of the Zhijin phosphorites plot in the IIIb domain, which belongs to the real Ce anomaly range (Fig 9a), it appears to reflect the phosphorites were formed under oxidic conditions.

However, many studies believe that the Ce anomalies in francolite may also involve REY fractionation during diagenetic uptake and relate to porewater REY chemistry, which cannot effectively constrain the redox environment [38–40] If the poor correlation between Ce/Ce* values and Y/Ho ratios in phosphorites indicates that this principle is also applicable to phosphorites [40]. There is a poor correlation between Ce/Ce* values and Y/Ho ratio ($R^2 = 0.06$, p>0.05) (Fig 9b), indicating that the Ce anomaly of Zhijin phosphorites may not only be affected by the REDOX state of seawater, but also may be caused by REY fractionation during diagenetic absorption. Therefore, the Ce anomaly cannot be completely used to limit the REDOX history of Zhijin phosphorites.

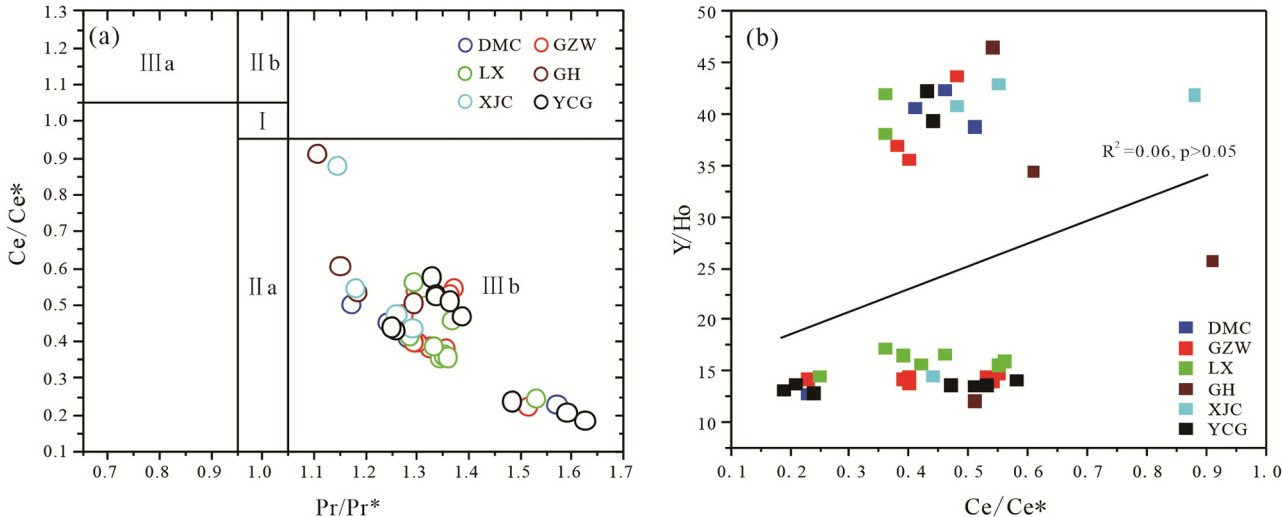

**Fig 9.** Position of the data points on the bivariate plot of (a) Pr/Pr* vs Ce/Ce* [25] and (b) Y/Ho vs Ce/Ce*. Domain I—no Ce anomalies and no La anomalies. Domain IIa—positive La anomalies and no Ce anomalies. Domain IIIb—negative La anomalies and no Ce anomalies. Domain IIIa—positive Ce anomalies. Domain IIIb—negative Ce anomalies.

**Trace elements and redox conditions.** The solubilities of V, Cr, Ni, Co, and other elements in seawater vary greatly. This easily results in differentiation and is reflected in the sediments [41, 42]. Therefore, many researchers have studied the redox environment of the ancient ocean based on the characteristics of these trace elements in marine sedimentary rocks [41–46].

V and Cr are soluble in water under oxic conditions, and they tend to be enriched in sediments deposited under reducing environment [47]. However, the reduction of V occurs in the lower part of the denitrification boundary, while the reduction of Cr occurs in the upper part of the boundary [47]. Therefore, the V/Cr ratio can be used as a parameter to determine the paleo-marine redox environment [41, 48, 49]. Generally, V/Cr < 2.00 indicates an oxygen-rich environment, 2.00 < V/Cr < 4.25 indicate a sub-oxygen-rich environment, and V/Cr >4.25 indicates an anoxic environment [41]. Co is dissolved in seawater in the form of $Co^{2+}$ under oxic conditions, or it forms complexes with humic acid, while insoluble CoS is formed under anoxic conditions [45]. Ni exists in oxidizing marine environments as $Ni^{2+}$, $NiCl^+$, and soluble $NiCO_3$, and it also forms complexes with humic acid [45, 50]. Under strongly reducing conditions, Ni forms insoluble NiS compounds [42]. Both Ni and Co are enriched in sediments formed in reducing environments, and the differences in their geochemical behaviors result in a certain correlation between their contents, Ni/Co < 5.00 indicates an oxic environment, 5.00< Ni/Co <7.00 indicate a sub-oxygen-rich environment, and Ni/Co > 7.00 indicates an oxygen-poor or anoxic environment [41].

The binary diagrams of V/Cr vs Mo and Ni/Co vs Mo for the lower Cambrian Zhijin phosphorites illustrates the consistency of the redox state of the sedimentary seawater (Fig 10a and 10b). Some samples from the XJC and GZW areas exhibit the characteristics of anoxic seawater, while the other samples exhibit the characteristics of oxygen-rich or sub-oxygen-rich seawater. The XJC and GZW phosphorite samples exhibit inconsistent redox conditions, which may be due to the special properties of the carbonate, which limit the expression of the trace elements [51], or the input of terrigenous detrital material in the deposition process [24]. The V/Cr and Ni/Co ratios of the lower Cambrian Zhijin phosphorite samples generally exhibit the characteristics of oxygen-containing seawater. This oxygen-rich environment provided the dynamic conditions required for the reproduction of organisms and the decomposition of

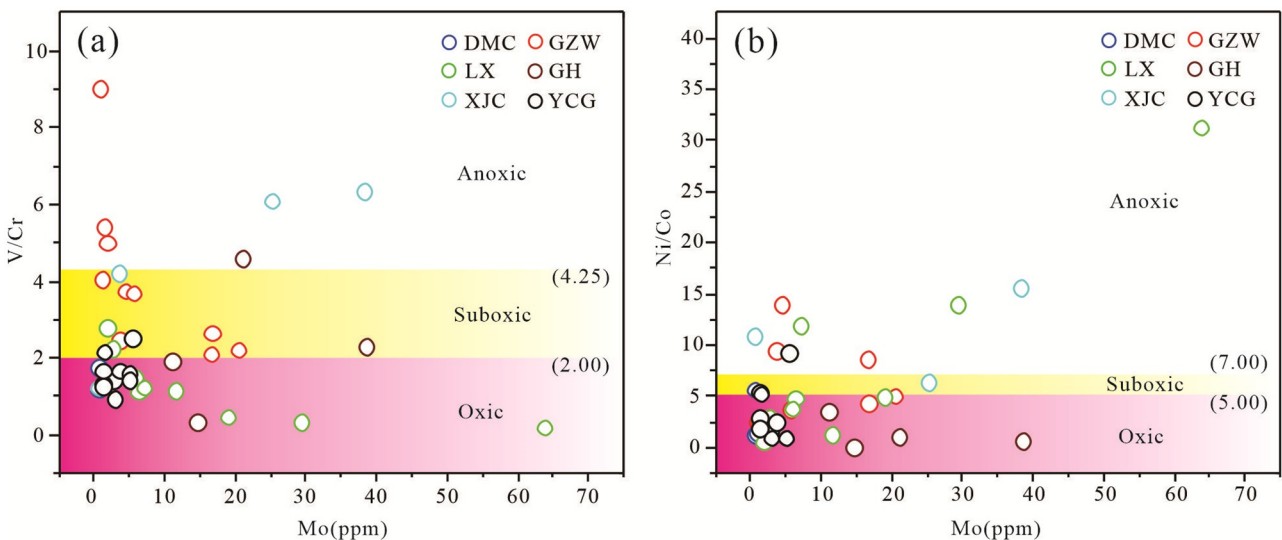

**Fig 10.** The binary diagrams of (a) V/Cr vs Mo and (b) Ni/Co vs Mo showing the redox conditions of the sedimentary seawater (after [41]).

biological remains, which were conducive to the participation of organisms and to their subsequent mineralization.

**Diagenetic weathering and post-diagenesis reworking.** The Y anomaly ($Y/Y^*$) and La/Nd ratios exhibit positive anomalies in seawater [27, 52], and they are not affected by redox changes; however, in sediments, these anomalies may decrease with increasing diagenesis [53]. Therefore, the $Y/Y^*$ and La/Nd ratios are valuable indicators for estimating post-depositional changes related to variations in the compositions of the circulating fluids [23, 27, 37]. Shields and Stille [53] reported that the $Y/Y^*$ and La/Nd ratios of modern seawater are 1.5–2.3 and 0.8–1.3, respectively. In the Zhijin phosphorites, the $Y/Y^*$ and La/Nd ratios of the phosphatic samples are 0.44–1.77 and 0.36–1.65, respectively. The $Y/Y^*$ vs La/Nd bivariate plot (Fig 11a) [27, 37, 53] shows that most of the values of both parameters for the phosphatic rocks are in the lower part of or lower than the seawater range. This also confirms that the Zhijin phosphorites experienced strong diagenesis, and the influence of weathering was weak.

Second, the Y anomaly ($Y/Y^*$) and $(La/Sm)_N$ ratios are important geochemical indexes for studying the intensity of weathering during the evolution of phosphorus blocks, and weathering results in a strong positive correlation between these two parameters [15, 37, 53]. The $Y/Y^*$ vs $(La/Sm)_N$ bivariate plot (Fig 11b) indicates a weak positive correlation ($R^2 = 0.43$, $p<0.05$) between $Y/Y^*$ and $(La/Sm)_N$ in the Zhijin phosphorites. Leaching experiments on phosphatic shales have revealed that MREE are leached preferentially compared to other REE [54], suggesting that the weathered remnant samples should exhibit MREE depletion [15, 37]. However, in the Zhijin phosphorites, the $(MREE/LREE)_N$ and $(MREE/HREE)_N$ values are both greater than 1, indicating that the MREE are more enriched than the LREE and HREE (S3 Table). The weak positive correlation between $Y/Y^*$ and $(La/Sm)_N$ and the MREE enrichment indicate that weathering did not play an important role in the REE distribution patterns of the Zhijin phosphorites.

The REY contents and δCe, δEu, and $(Dy/Sm)_N$ values of phosphorites are affected by post-diagenesis reworking, leading to good correlations, while δCe and REYs correlations of the original phosphorites are poor [53]. If the $(La/Sm)_N$ ratio is not correlated with δCe and the $(La/Sm)_N$ ratio is greater than 0.35, the Ce anomalies in the apatite represent the original

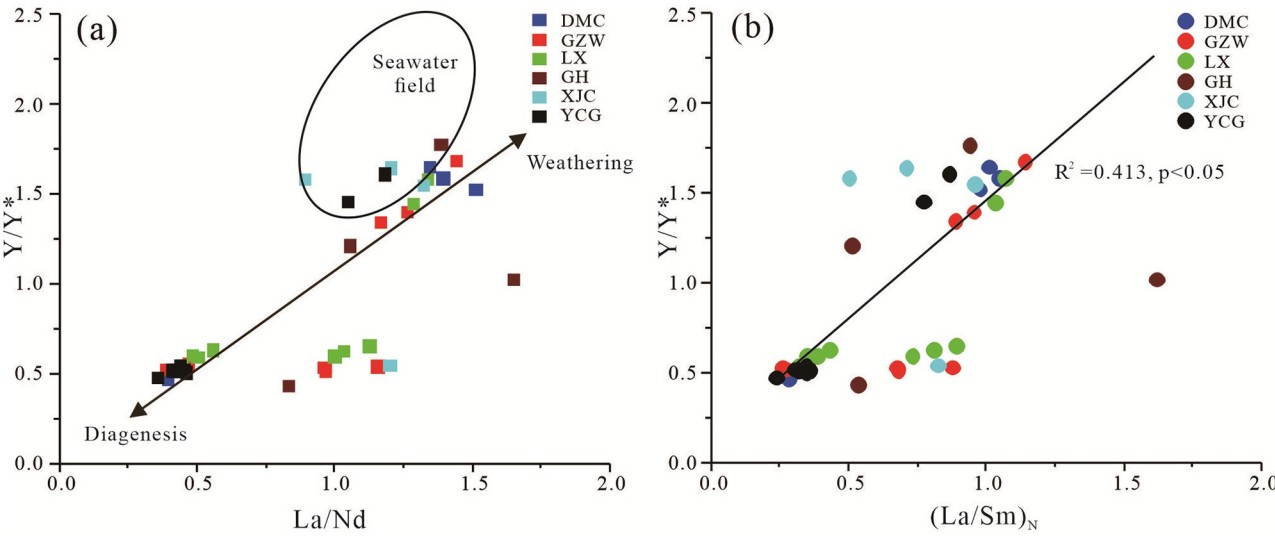

**Fig 11.** Bivariate plots of (a) $Y/Y^*$ vs La/Nd and (b) $Y/Y^*$ vs $(La/Sm)_N$ for the phosphatic rocks of the lower Cambrian phosphorites in Zhijin area. The seawater field is from [27].

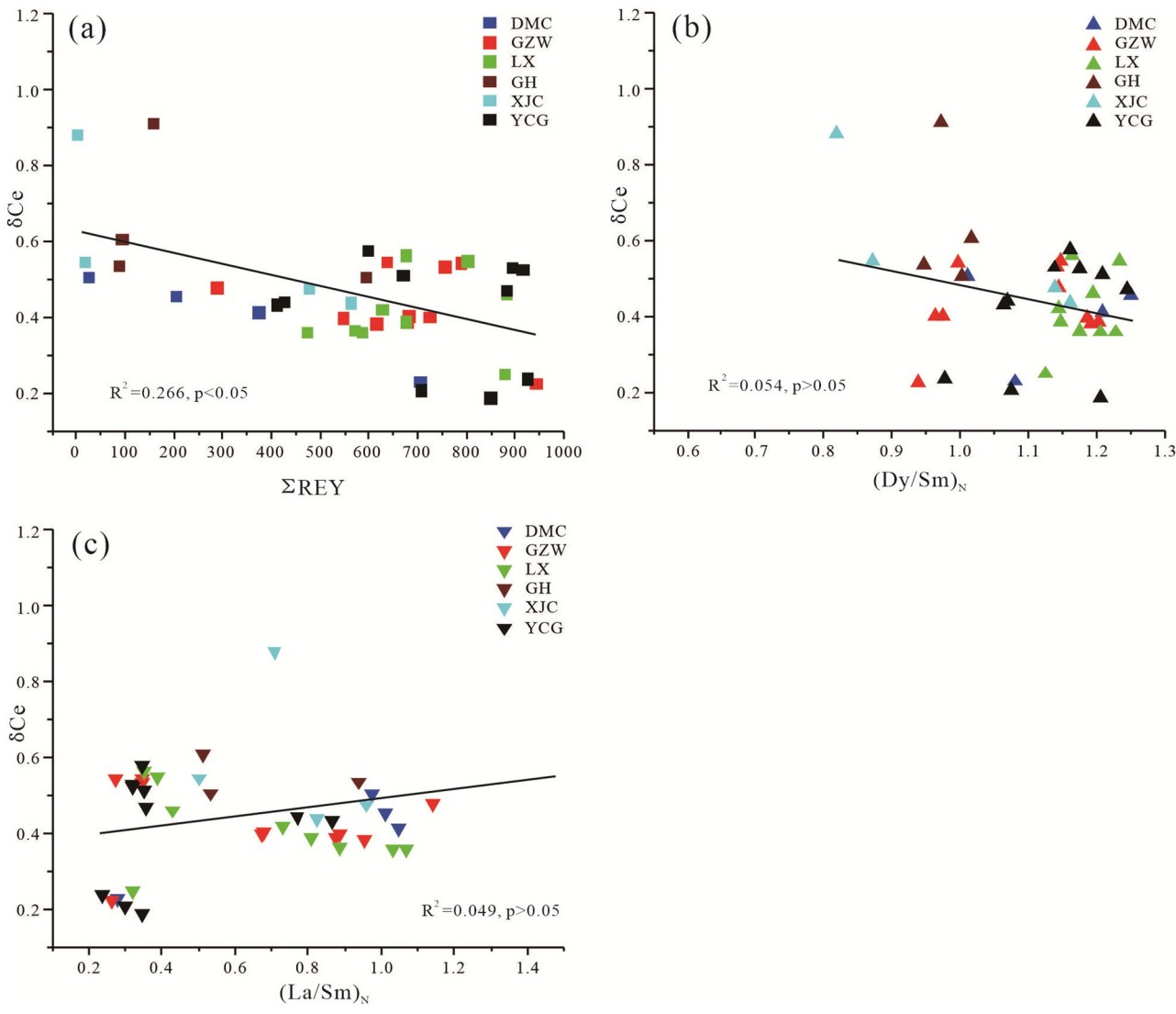

**Fig 12.** Scatter plots of (a) δCe vs ΣREY, (b) δCe vs $(Dy/Sm)_N$, and (c) δCe vs $(La/Sm)_N$.

signature of the seawater [36]. However, we found that there were no correlations between δCe and ΣREY (Fig 12a), δCe and $(Dy/Sm)_N$ (Fig 12b), and δCe and $(La/Sm)_N$ (Fig 12c) for the Zhijin phosphorites, and most of the $(La/Sm)_N$ values were greater than 0.35 (Fig 12c). This indicates that the Ce anomalies of the Zhijin phosphorites are the original signature of the seawater, and the entire REYs series was only slightly affected by post-diagenetic modification.

## Source of diagenetic material

**Consistency of diagenetic material source.** Correlation analysis of the elements in the lower Cambrian phosphorus block in the Zhijin area revealed that there is a positive correlation between the $P_2O_5$ and CaO contents, with a correlation coefficient of 0.62 (Fig 13a). There are positive correlations between the $P_2O_5$ content and the CaO and ΣREY contents, with correlation coefficients of 0.83 and 0.33, respectively (Fig 13b and 13c). This shows that the REE and phosphorus had the same material source. In addition, we normalized the REY

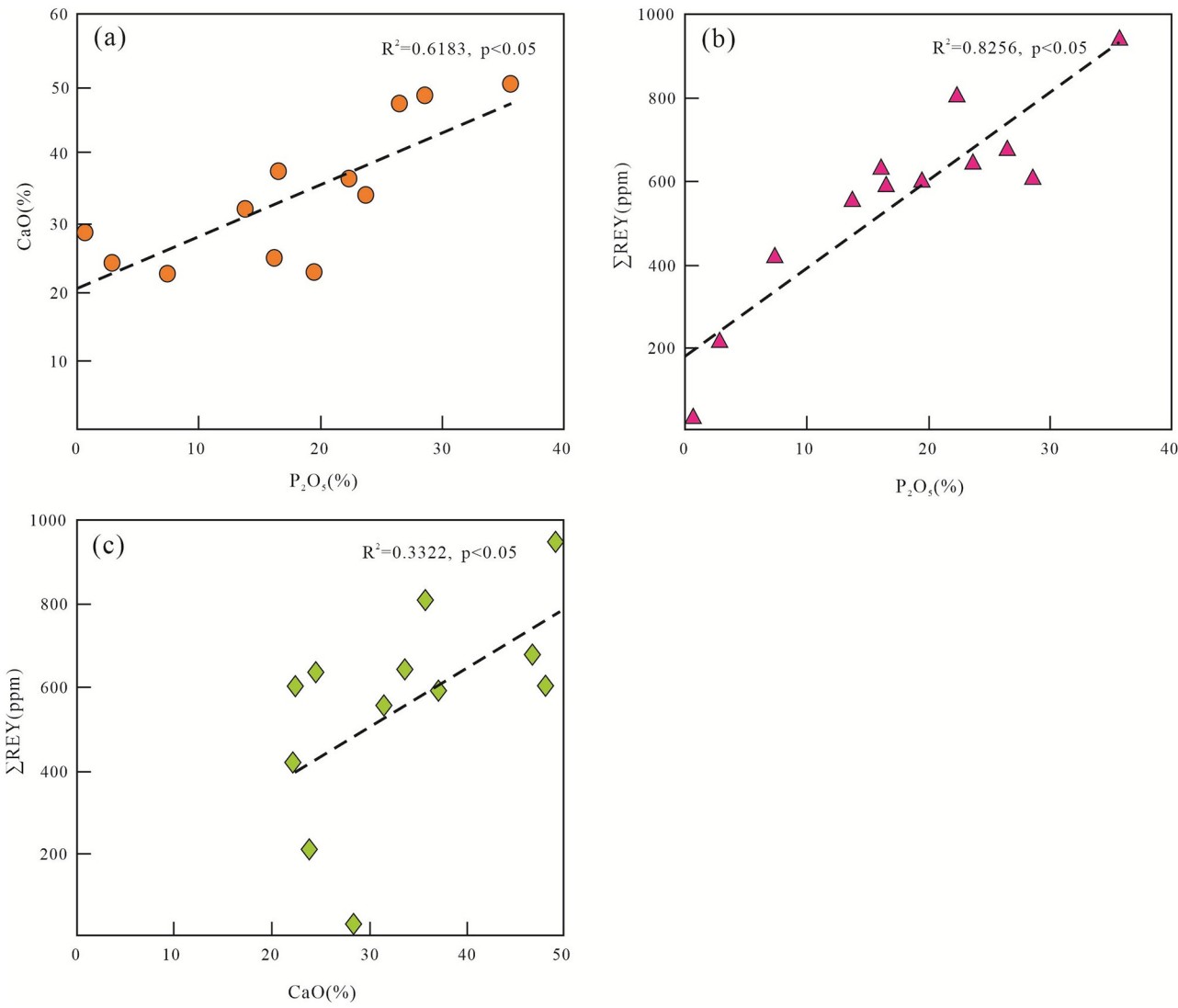

**Fig 13.** Binary diagrams for selected elements showing correlations between: (a) CaO vs P2O5; (b) ΣREE vs P2O5; and (c) ΣREY vs CaO in the phosphorites of Zhijin area.

data of each profile of the Zhijin phosphorites to PAAS (McLennan, 1989) and conducted REY correlation analysis (Fig 14). It was found that the Pr–Lu segment had a strong correlation, and there were positive Ce, La, and Y anomalies, but the REYs still exhibited strong overall homology. The positive anomaly and weak correlations of Y may be because the source of the ore-forming materials was complex and was not a single source [23].

Additionally, the REE distribution patterns for similar sedimentary environments have similar shapes [55]. The REE data were normalized to PAAS [26], and the REE distribution pattern of the Zhijin phosphorite in each profile was obtained. The YCG, GZW, and LX profiles have similar patterns (Fig 15b, 15c and 15f), while the GS and DMC profiles form another group with similar patterns (Fig 15a and 15d). These results indicate that the REE distribution patterns tend to be consistent for similar or identical sedimentary microfacies environment. The rationality of the sedimentary microfacies zone can also be inferred from these characteristics.

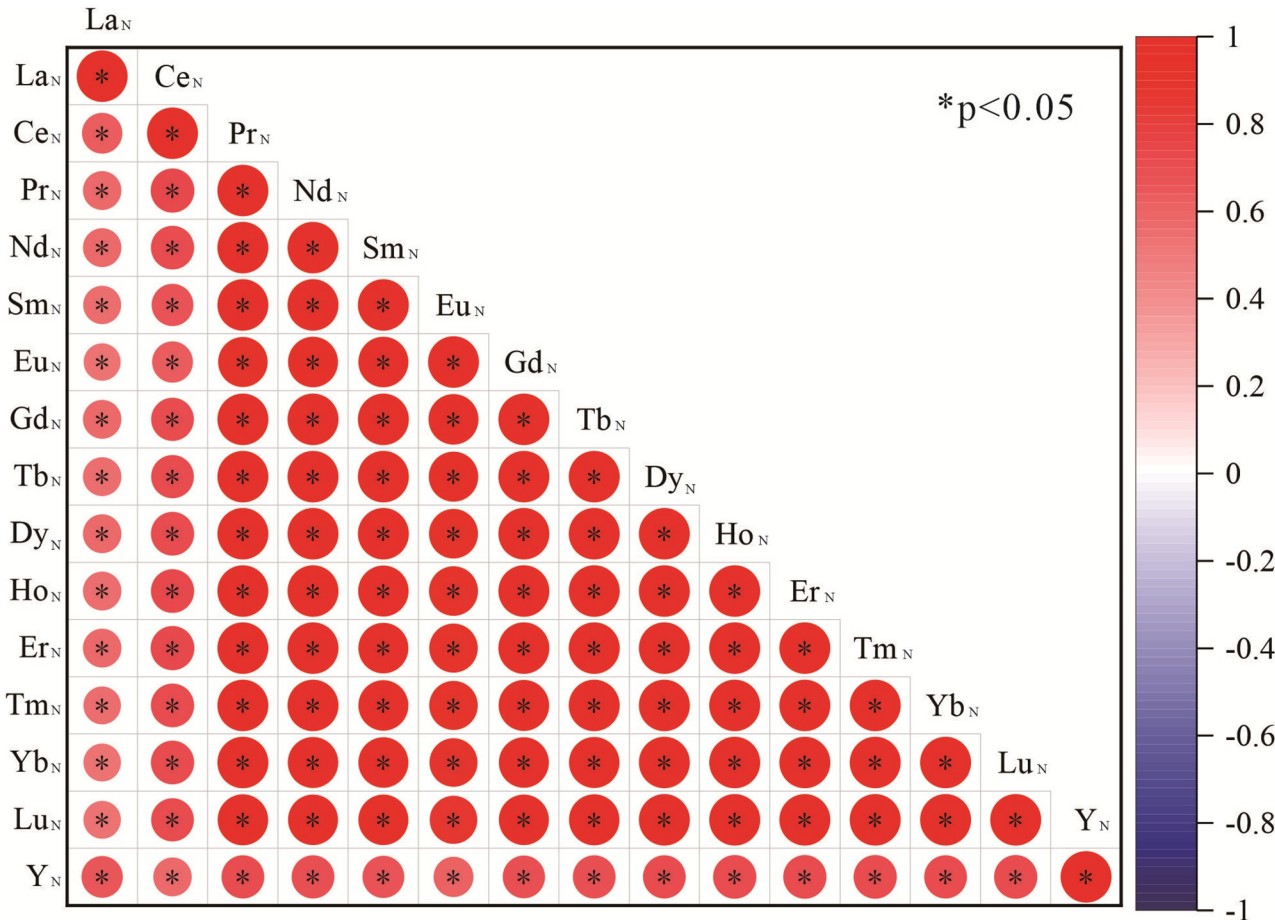

**Fig 14. Spearman correlation analysis of the REYs of the phosphorites in Zhijin area.**

**Seafloor hydrothermal activity.** Positive Eu anomalies ($\delta$Eu) are usually used as indicators of a reducing seafloor hydrothermal supply [37, 56]. These anomalies may have been caused by the presence of a high temperature reducing hydrothermal influence during the deposition process, which enabled the stable existence of $Eu^{2+}$ and resulted in the positive Eu anomalies [57, 58]. Some researchers believe that the occurrence of positive Eu anomalies is connected to organic matter [59]. However, in normal seawater sediments, there are generally no obvious Eu anomalies [58]. The $\delta$Eu values of the Zhijin phosphorites range from 0.92 to 3.83, with a median of 1.12 and an average of 1.26, indicating weak positive Eu anomalies (Fig 16a). This indicates that the Zhijin phosphorites were affected by seafloor hydrothermal activity, but not significantly.

In addition, the As and Sb contents can be used to distinguish seafloor hydrothermal sedimentary activity from normal sedimentary activity [60]. The average As and Sb contents of seafloor hydrothermal sedimentary deposits are greater than 100 ppm and 7 ppm, respectively. The average As and Sb contents of normal seawater deposits are 10 ppm and 2–3 ppm, respectively [60]. The Ag concentrations of modern Pacific mid-ridge seafloor hydrothermal sediments range from 5 to 186 ppm (average of 37 ppm) [61]. The Ag content can be used to judge whether the deposition was affected by hydrothermal activity [23]. The Ag contents of the lower Cambrian Zhijin phosphorites from 0.02 to 27.5 ppm, with a median of 1.92 ppm and

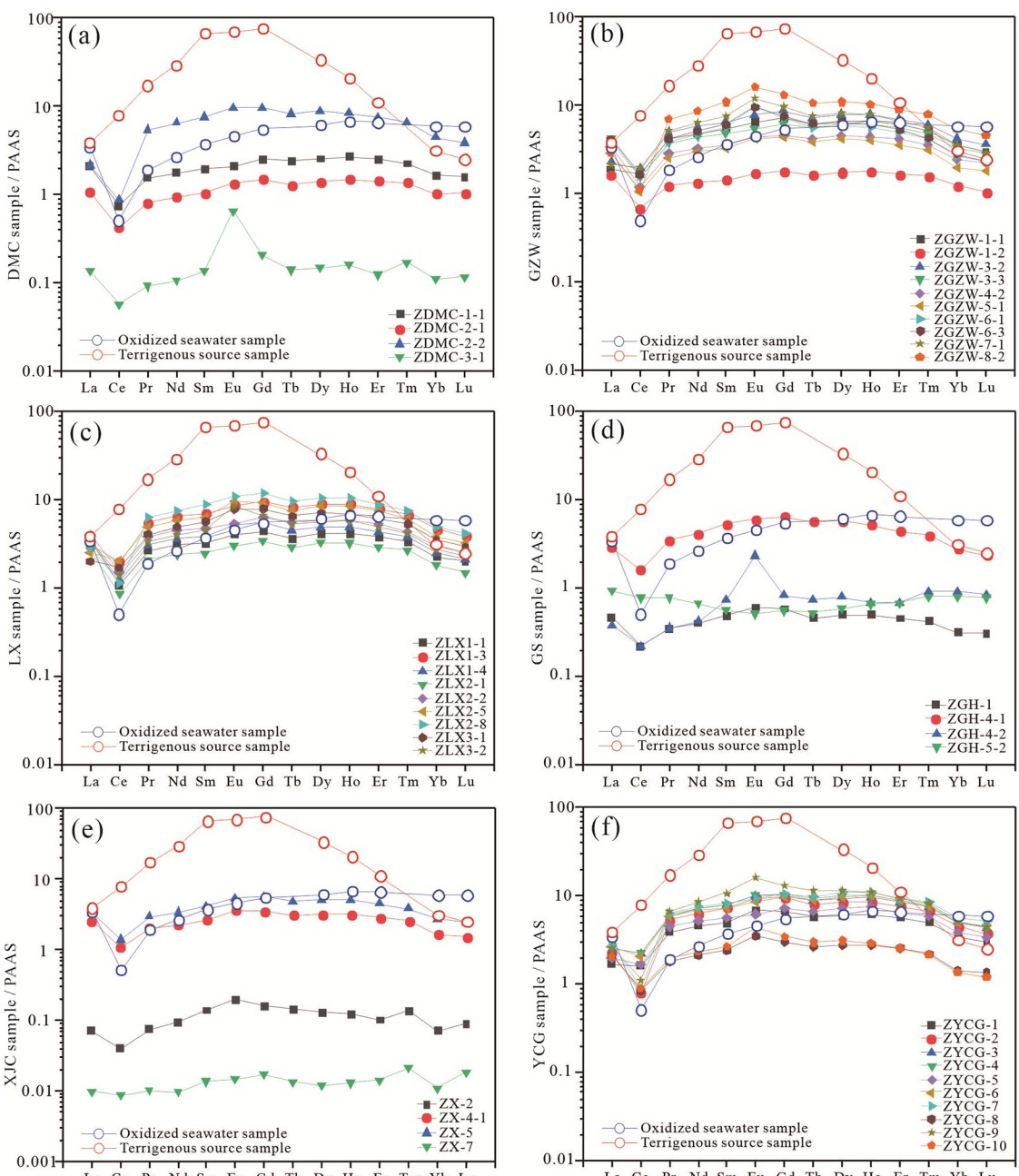

**Fig 15.** PAAS-normalized REE patterns of samples from the: (a) DMC location; (b) GZW location; (c) LX location; (d) GS location; (e) XJC location; (f) YCG location in the lower Cambrian Zhijin deposit.

an average of 4.31 ppm (Fig 16b). The As contents range from 0.11 to 156 ppm, with a median of 12.4 ppm and an average of 18.36 ppm (Fig 16c). The Sb contents range from 0.04 to 116.51 ppm, with a median of 9.85 ppm and an average of 20.43 ppm (Fig 16d). Based on the Ag and As contents, normal seawater deposition occurred; however, the Sb contents of some of the samples exhibit the characteristics of seafloor hydrothermal deposition. Based on the Ag, As, and Sb contents, i.e., the representative elements of hydrothermal activity, and the weak Eu positive anomaly, the lower Cambrian Zhijin phosphorites were slightly affected by a

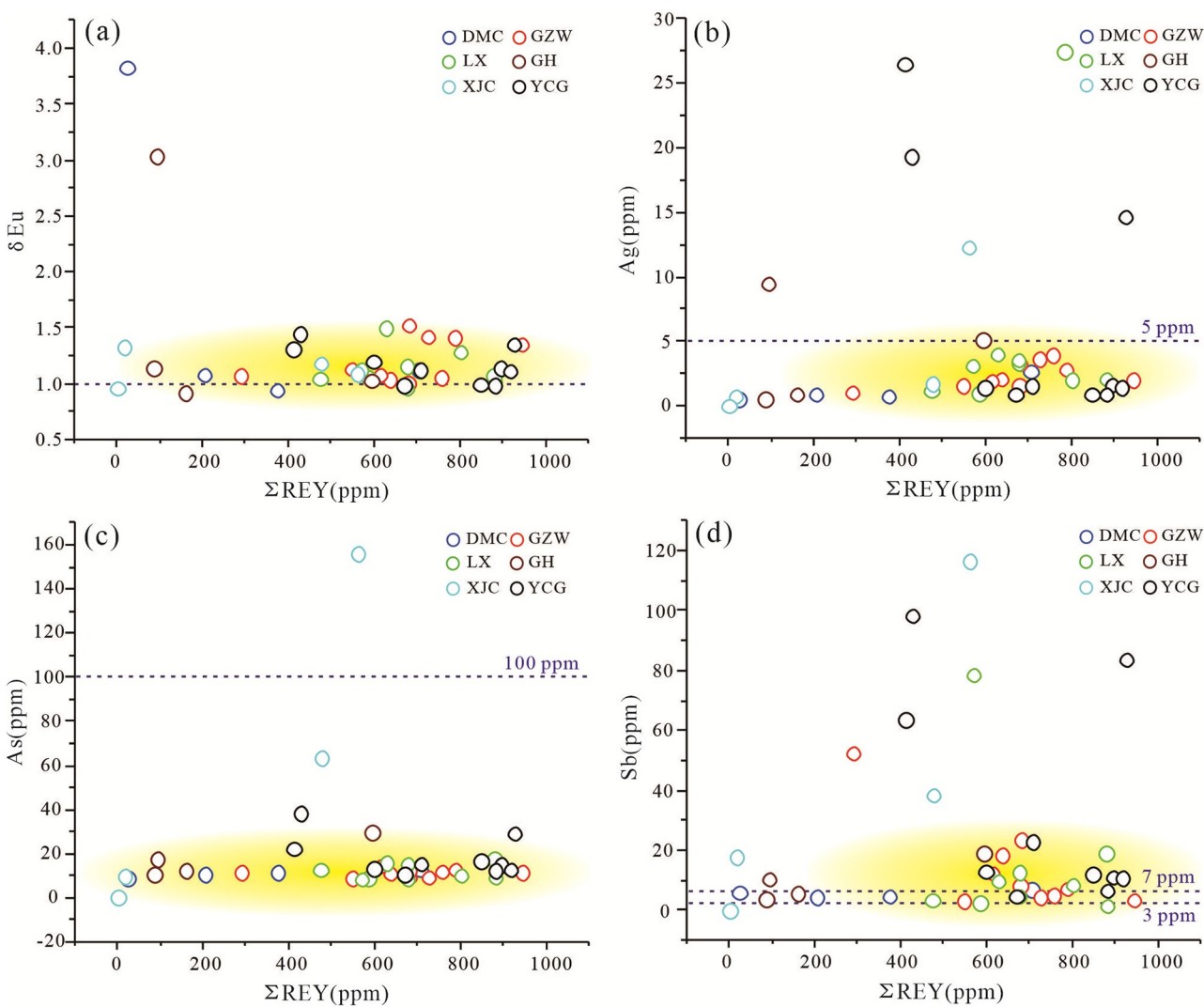

**Fig 16.** Bivariate plots of (a) δEu vs ΣREY, (b) Ag vs ΣREY, (c) As vs ΣREY, and (d) Sb vs ΣREY for the phosphatic rocks of the lower Cambrian phosphorites in Zhijin area.

high temperature hydrothermal fluid during normal deposition, but the submarine hydrothermal activity was not the direct replenishment source of the ore-forming materials.

## Source of terrestrial material

There is a difference in the Y/La ratios of seawater and upper continental crust during the deposition of phosphorus blocks, resulting in different degrees of La and Y absorption after the deposition and diagenesis of biological phosphorus blocks. Previous studies have mapped the Y/La ratios of seawater and the upper continental crust using the components produced by most modern and ancient biological apatite samples [62]. By comparing the Y/La ratios of the Zhijin phosphorites samples with the results of previous studies [62], it was found that the Y/La ratios indicate that some of the samples ploted in the seawater source range, and some of the samples plotted in the upper continental crust characteristics range, but none of the samples exhibited upper crustal characteristics (Fig 17a).

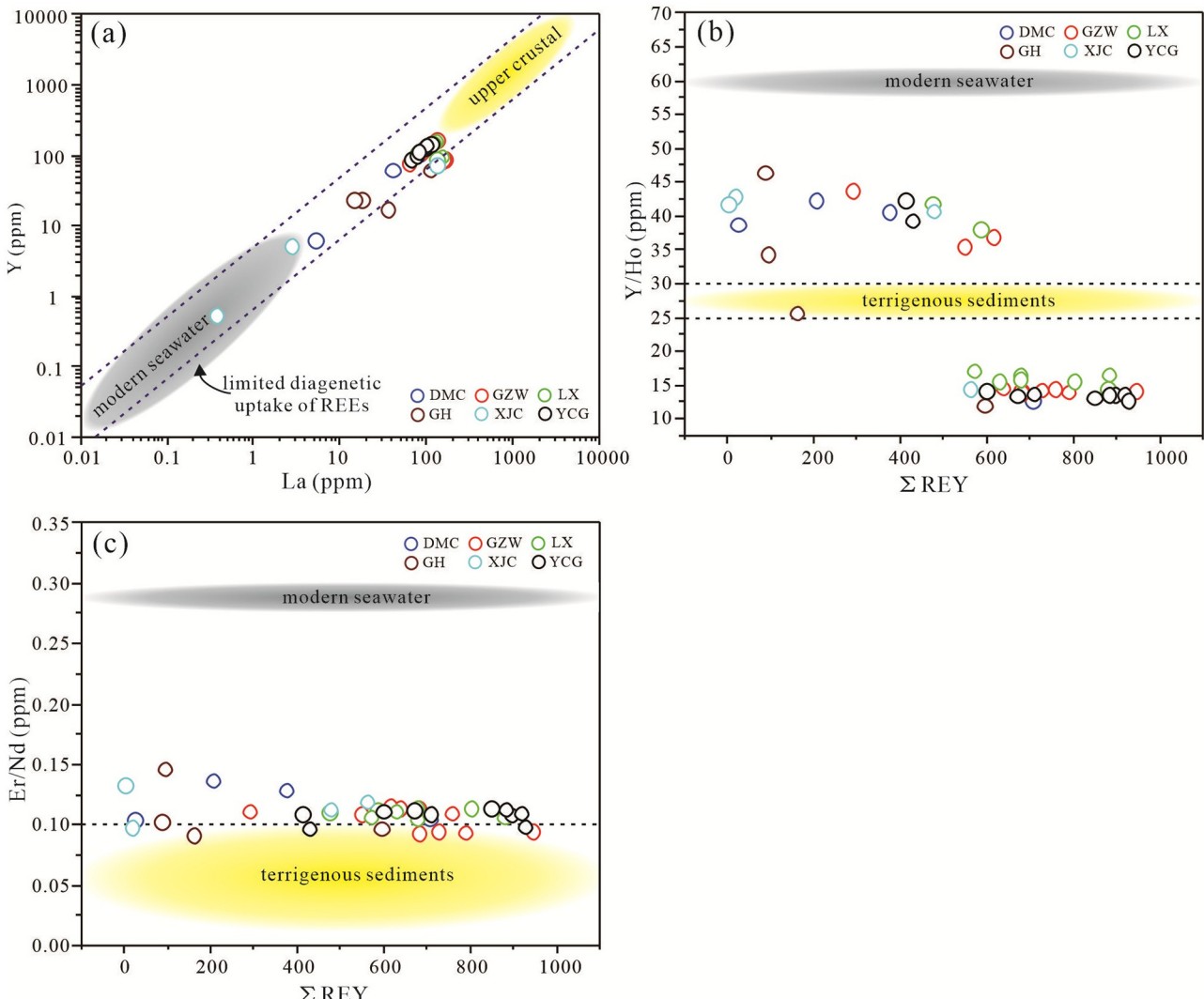

**Fig 17.** Bivariate plots of (a) Y vs La (base map adapt from [62], dashed diagonal lines show the weight ratio of Y/La in modern seawater (4.7; [63]) and upper continental crust (0.73; [64])); (b) Y/Ho vs ΣREY (reference data for seawater, land and debris come from [37]); and (c) Er/Nd vs ΣREY (Reference data for seawater, land and debris come from [37]) for the phosphatic rocks of the lower Cambrian phosphorites in Zhijin area.

The terrestrial origin of the REYs in the Zhijin phosphorites can be further evaluated using the Y/Ho ratio. Y and Ho have similar ionic radii in the modern ocean [52, 65], but Ho is removed from seawater about twice as rapidly as Y [52, 66, 67]. This process results in the Y/Ho ratios of seawater and terrigenous siliciclastics being approximately 60 and 25–30, respectively [23, 62, 64]. The Y/Ho ratios of the Zhijin phosphorites samples range from 12.87 to 46.53, with a median value of 15.65, and most of the samples are not within terrigenous range (Fig 17b). The Er/Nd ratio of normal seawater is approximately 0.27 [68], and clastic materials and diagenesis may promote Nd enrichment relative to Er, thus reducing the Er/Nd ratio to less than 0.1 [62, 69]. The Er/Nd ratios of the Zhijin phosphorites samples range from 0.09 to 0.15, with a median value of 0.11, and some of the samples with values of less than 0.1 exhibit terrigenous characteristics (Fig 17c). Based on the Y/La, Y/Ho, and Er/Nd ratios, the REE enrichment process of the Zhijin phosphorus block was affected by the input of terrigenous detrital material, but the effect was not significant.

**Source of oxidized seawater.**   The REE distribution patterns of sedimentary rocks can be used as an important indicator to determine the sedimentary environment [34, 35, 70, 71]. The REE distribution patterns (Fig 15a–15f) of the Zhijin phosphorite profiles exhibit similar characteristics, i.e, strong negative Ce anomalies, weak positive Eu anomalies, MREE enrichment, and HREE depletion.

Research on old and modern phosphatic deposits has demonstrated that seawater-like REE patterns are distinguished by enrichment of HREE and negative Ce anomalies [25, 70]. By comparing the REE distribution patterns of the Zhijin phosphorite samples with those of typical oxic seawater samples [72] and terrigenous source samples [72], it was found that the REE distribution patterns of the Zhijin phosphorites exhibit characteristics similar to those of oxidative seawater in the La–Ho section, with strong negative Ce anomalies. However, the Er–Lu segment exhibits characteristics similar to those of terrestrial materials, with HREE depletion (Fig 15a–15f). Because the Zhijin phosphorites were not affected by diagenetic weathering and post-diagenesis reworking, they retain the original sedimentary characteristics, and submarine hydrothermal activity was not the direct supply source of the diagenetic materials. Based on the distribution characteristics of the REE, the diagenetic materials were mainly from oxidized seawater accompanied by weak terrigenous superposition. The oxidized seawater not only transported oxygen and nutrients for the lower Cambrian Zhinjin small shell biota, which promoted the flourishing of organisms, but it also provided an important material basis for diagenesis and mineralization.

## Biochemical action

According to the major element data, the phosphorus contents of the bioclastic phosphorites containing SSFs at the bottom of the Zhijin phosphorite-bearing series are higher than those of the siliceous phosphorites at the top. In particular, in the LX, GS, and GZW phosphorite-bearing series, the $P_2O_5$ contents are significantly higher. This indicates that the formation of the lower Cambrian phosphorite in Zhijin was influenced by organisms. According to the previous analysis, the sea water was always in an oxygen-rich state during the deposition of the Zhijin phosphorite block. The rising ocean currents supplied a large amount of phosphorus, and the phosphorite recorded the geochemical characteristics of hydrothermal deposition, which are the most suitable environmental conditions for the growth of small shell animals. During the Ediacara–Cambrian transition, the biodiversity was maintained, but the level was at least 25% of the present atmospheric level [73]. The oxidation background was not only important for oxygen and nutrient inputs (such as phosphorus enrichment), but it also provided an important material basis for diagenetic mineralization when the Cambrian biological explosion was triggered.

The enrichment of V and Ni is closely related to biological activities. The enrichment of V is affected by plankton and algae, while the enrichment of Ni is greatly affected by the activities of nearshore organisms. Therefore, the V/Ni ratio can reflect the depth of the seawater in which the sediments were deposited [74]. Except for the YCG area, the V/Ni ratios of the Zhijin phosphorite samples were greater than 0.5 (Fig 18a), and the V contents were higher during the deposition process. This was also related to the enrichment of Hyolitha, Zhijinites, and other zooplankton and algal activities in the phosphorites.

Phosphorites rich in SSFs tend to have high REE contents, which are much higher than the average REE abundance of the Earth's crust (178.00 ppm). Through correlation analysis of the $P_2O_5$ and CaO contents and $\Sigma$REE contents of the samples (Fig 13a–13c), it was found that there are positive correlations between the $P_2O_5$ and CaO contents and the $\Sigma$REE content, with correlation coefficients of 0.83 and 0.33, respectively. The different distribution coefficients of

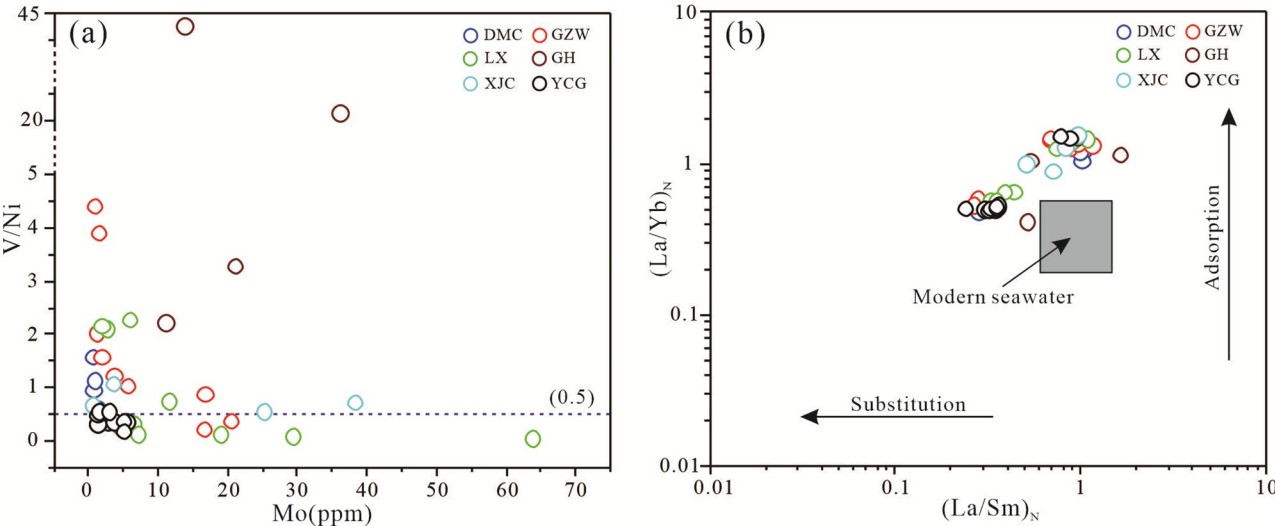

**Fig 18.** Bivariate plots of (a) V/Ni vs Mo and (b) (La/Yb)$_N$ vs (La/Sm)$_N$ (base map adapt from [75]) for the phosphatic rocks of the lower Cambrian phosphorites in Zhijin area.

the rare earth elements in apatite and seawater suggest various mechanisms, including adsorption and substitution [37]. According to the (La/Yb)$_N$ vs (La/Sm)$_N$ diagram (Fig 18b), the different mechanisms leading to the entry of rare earth elements into the apatite lattice can be distinguished [75]. For the Zhijin phosphorites samples, both ratios exhibit wide variations and a strong positive covariation, with (La/Yb)$_N$ values ranging from 0.41 to 1.55 and (La/Sm)$_N$ valeus ranging from 0.24 to 1.62 (Fig 18b), which are not within the range for modern seawater. The (La/Yb)$_N$ vs (La/Sm)$_N$ diagram shows that the adsorption and substitution mechanisms of carbonate-fluorapatite played an important role in the enrichment of REE during the diagenesis of the Zhijin phosphorites.

The main composition of the SSFs in the Zhijin phosphorites is carbonate-fluorapatite (Fig 5a–5d), which indicates that the shells absorbed a large amount of phosphorus to form bio-collophanite. The cementation material of some of the collophanite samples does not contain phosphorus, forming biophosphorus block deposits. The REE distribution of the ancient ocean was largely preserved by organisms due to the adsorption of marine organisms in the Cambrian [20].

Zhang et al. [8] argued that because of the seawater microbes in the lower Cambrian system, the decomposition of algae and other organisms occurred after their death, which increased the solubility of phosphorus in the water (e.g., increased carbonic acid, $CO_2$, ammonia), leading to the formation of phosphor via chemical deposition. After modification, the biological debris was mineralized with collophane, resulting in La, Nd, and Y being further enriched, with isomorphic displacement within the apatite lattice [8, 76, 77], resulting in REE enrichment. In general, carbonate-fluorapatite is the dominant composition of the SSFs in the Zhijin phosphorites (Fig 5a–5d), and the $Ca^{2+}$ in the carbonate-fluorapatite has an ionic radius similar to those of some REE [8, 78], which indirectly indicates that biological or biochemical activities played an important role in the genesis of the rare earth phosphate deposits in the Zhijin area.

## Metallogenic mechanism of the phosphorite block

Large-scale marine sedimentary phosphate deposits were formed in the Late Sinian Doushantuo stage and the Early Cambrian Meishucun stage on the Yangtze platform during two

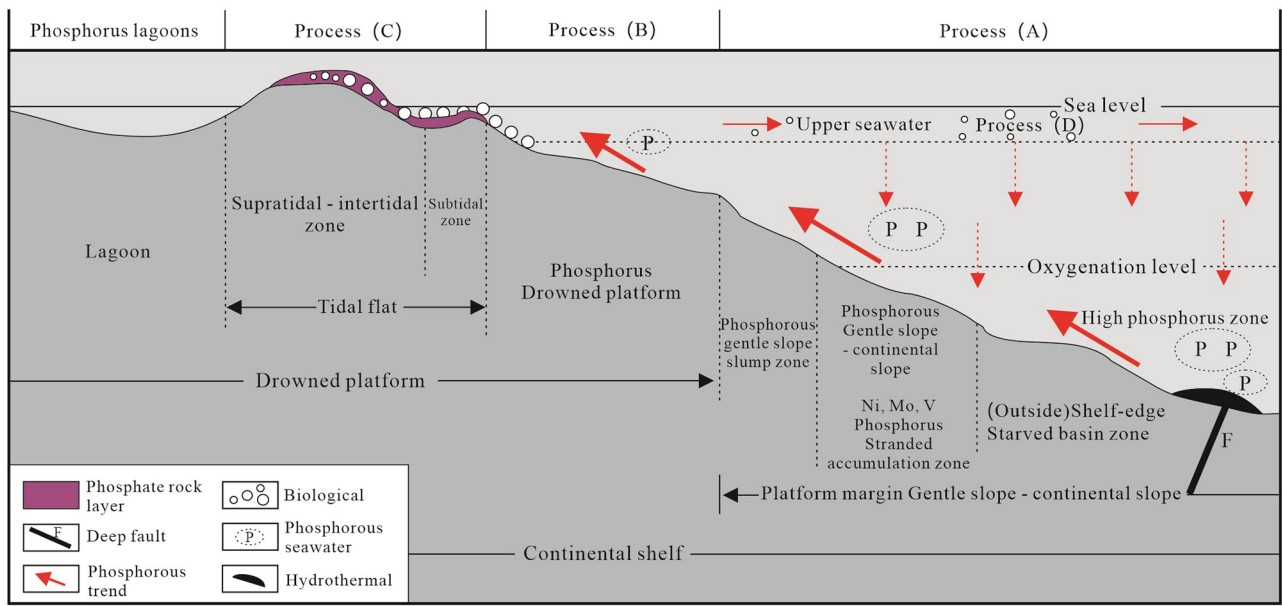

**Fig 19. Sedimentary model map of the phosphorite in the lower Cambrian Zhijin deposit.**

important stages of phosphorus formation [21, 22–24, 40, 78]. The phosphorus-bearing rock series in Zhijin was formed during the Meishucun period in the lower Cambrian [20, 40] and the formation of these phosphate deposits mainly included three factors: phosphorus source, migration, and pre-precipitation concentration [10, 23, 24, 79]. Based on the results obtained in this study and previous research data, the formation of the Zhijin phosphorous block has been summarized as follows (Fig 19).

Process A. Platform margin: The rising ocean currents transported abundant phosphorus and REE containing materials from the deep sea due to the large-scale transgression during the Meishucun period, and these currents reached the platform margin [20, 21, 24, 78]. Silt and other impurities in the high-phosphorus seawater were deposited first, resulting in a rapid increase in the P and Ca contents of the seawater. The temperature of the sea water increased accordingly, and the water also changed from a reducing environment to an oxidizing environment, so the organisms flourished in large quantities [79]. Additionally, the biochemical activities lead to the precipitation of phosphate. At this time, the water at the edge of the platform was deep, which was not conducive to the deposition and mineralization of phosphorus, so phosphate deposits seldom formed during this process (Fig 20a) [10, 79].

Process B. Shallow sedimentary environment: After process A, the transgression continued and the seawater reached the shoal area, which flooded the platform, making the water shallower, which was conducive to the deposition of phosphorus in the seawater [79]. During process B, the $O_2$ content increased, the sunlight was sufficient, and the marine organisms flourished [8]. Eventually, under the joint influence of the biological and chemical actions, the phosphate in the seawater reached the state of supersaturation, and phosphate deposits formed (Fig 20b).

Process C. Tidal flat deposits: The seawater reached the tidal flat sedimentary zone of the shallow sea and formed an oxidizing water environment. This was most suitable for the deposition of the phosphorite block, and the Zhijin phosphate deposit was formed in this stage (Fig 20) [40, 78, 79]. The phosphorus-rich water supplied by the upwelling current reached the

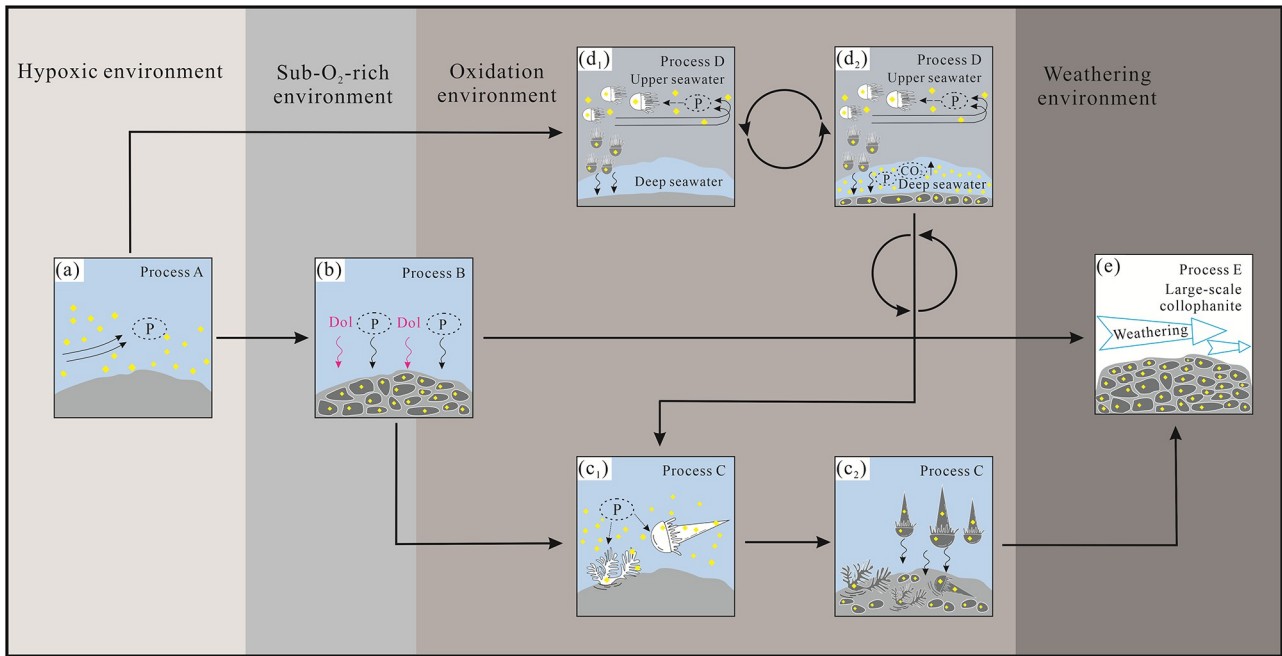

**Fig 20.** Bioconcentration of phosphorus model map of lower Cambrian Zhijin deposit: (a) Rising ocean currents supplies large amounts of phosphorus to the platform margin; (b) phosphate in the seawater reaches supersaturation, and phosphatic or dolomitic material precipitates; ($c_1$) small shell fauna prosper in large numbers and absorb phosphorus to form their shells; ($c_2$) the organisms die and their phosphorus-rich shells accumulate; and shells continue to adsorb phosphorus and the solubility of phosphorus in water is enhanced, promoting phosphatic accumulation via chemical precipitation; ($d_1$) plankton carry phosphorus from the upper seawater into the deep seawater; ($d_2$) the increase in the $CO_2$ content of the deep seawater facilitates the dissolution of phosphorus in the deep seawater; (e) large-scale collophanite is formed and weathered.

shallow sea area where benthic algae mainly developed [79]. As the phosphorus content of the water increased, owing to the suitable temperature, hydrodynamics, and $O_2$ content of the sea water, small shell fauna prospered and absorbed phosphorus to form their shells [4, 6, 11]. In addition, the adsorption of organisms also promoted the decomposition and precipitation of phosphorus from the phosphorus-containing organic matter (Fig 20$c_1$) [10]. With the flourishing biological activity, the physicochemical properties of the shallow sea area changed, becoming rich in a large number of phosphorus accumulating biogenic organisms, which were buried after death [3–5]. In addition, phosphorus adsorption by shells continued to play an important role, and the water soluble phosphorus enhancement promoted the accumulation of phosphatic material via chemical precipitation (Fig 20$c_2$) [8, 10]. Phosphorus was enriched and mineralized, and collophanite mineralization occurred in shallow sea water, forming a large-scale, high $P_2O_5$ grade phosphate deposit (Fig 20e) [10, 40, 78, 79].

Process D. Upper seawater environment: The phosphorus in the seawater was absorbed by the abundant plankton, and their death transported part of the phosphorus into the deep water (Fig 20$d_1$) [10, 24]. The $CO_2$ content of the deep seawater increased, facilitating the dissolution of phosphorus in the deep seawater (Fig 20$d_2$) [8]. The phosphorus content of the water increased rapidly, and the seawater biomass continued to flourish. This process continued throughout the transgression phase [19, 80].

## Conclusions

1. The sedimentary environment of the Zhijin phosphorus-bearing rock series changed from a low energy environment (subtidal-intertidal zone) to a high energy environment

(supratidal zone), and the sea water depth decreased until a biological shoal was formed in the XJC area. The rising ocean currents transported abundant phosphorus and REE containing materials from the deep sea during the large-scale transgression in the Meishucun period and reached the platform margin. The phosphorite was subjected to the influence of a high temperature hydrothermal fluid during normal deposition.

2. The different sedimentary environments of the different phosphorus-bearing rock series lead to the obvious differences in their biofossil assemblages. In the high-energy environment, the species were abundant, especially Hyolithes, which exhibited large individual development and lip development. In the low-energy environment, benthic algae mainly developed. In particular, it was found that the Hyolithes species had a strong environmental adaptability and appeared in various sedimentary environments in different individual forms.

3. The REE content of the Zhijin biophosphate is very high, and this REE enrichment may be related to the activities of organisms. The calculation results of Ce anomaly, V/Cr ratio and Ni/Co ratio demonstrate that the lower Cambrian Zhijin phosphorites were mainly formed in oxidic environment. This oxygen-rich environment is conducive to the participation of organisms and to their subsequent mineralization.

4. The high phosphorus content of the seawater in the lower Cambrian was related to the outbreak of small shelled animals in the same period. The biological activities changed the physicochemical properties of the seawater, such as pH, $CO_2$ concentration, and pressure, and formed favorable conditions for phosphate deposition. The biological activities contributed significantly to the phosphate formation in the Meishucun stage.

## Supporting information

**S1 Table. Concentrations of major elements (wt%) of the Zhijin phosphorites samples in the lower Cambrian strata.**
(DOCX)

**S2 Table. Concentrations of trace and rare earth elements (ppm) of the Zhijin phosphorites samples in the lower Cambrian strata.**
(DOCX)

**S3 Table. Calculated values of some geochemical parameters of the Zhijin phosphorites samples in the lower Cambrian strata.**
(DOCX)

## Acknowledgments

We thank LetPub (www.letpub.com) for its linguistic assistance during the preparation of this manuscript.

## Author Contributions

**Conceptualization:** Lei Gao.

**Investigation:** Lei Gao.

**Methodology:** Lei Gao.

**Supervision:** Ruidong Yang.

**Writing – original draft:** Lei Gao.

**Writing – review & editing:** Lei Gao, Tong Wu, Chaokun Luo, Hai Xu, Xinran Ni.

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
