## [Decision Letter · Decision Letter 0]

11 Oct 2022

PONE-D-22-21235Studies on geochemical characteristics and biomineralization of Cambrian phosphorites, Zhijin, Guizhou Province, ChinaPLOS ONE

Dear Dr. Yang,

Thank you for submitting your manuscript to PLOS ONE. I apologize for the long review process since I have had difficulty in finding appropriate reviewers for your work. After careful consideration, we feel that this manuscript is well-written with very good quality, but it does not fully meet PLOS ONE’s publication criteria as it currently stands. Therefore, we invite you to submit a revised version of the manuscript that addresses the points raised during the review process.

The reviewers suggest significant revisions on the introduction, methods, results and discussion parts of the manuscript, as well as modification of figures. I would recommend you considering the comments from the reviewers and provide a point-by-point response. Please note that the revised manuscript may be re-reviewed before considering for publication.

We look forward to receiving your revised manuscript.

Kind regards,

Ziming Yang, PhD

Academic Editor

PLOS ONE

Journal Requirements:

  Whilst you may use any professional scientific editing service of your choice, PLOS has partnered with both American Journal Experts (AJE) and Editage to provide discounted services to PLOS authors. Both organizations have experience helping authors meet PLOS guidelines and can provide language editing, translation, manuscript formatting, and figure formatting to ensure your manuscript meets our submission guidelines. To take advantage of our partnership with AJE, visit the AJE website (http://aje.com/go/plos) for a 15% discount off AJE services. To take advantage of our partnership with Editage, visit the Editage website (www.editage.com) and enter referral code PLOSEDIT for a 15% discount off Editage services. If the PLOS editorial team finds any language issues in text that either AJE or Editage has edited, the service provider will re-edit the text for free.

Additionally, in your manuscript, please provide additional information regarding the specimens used in your study. Ensure that you have reported specimen numbers and complete repository information, including museum name and geographic location. If permits were required, please ensure that you have provided details for all permits that were obtained, including the full name of the issuing authority, and add the following statement: 'All necessary permits were obtained for the described study, which complied with all relevant regulations.' If no permits were required, please include the following statement: 'No permits were required for the described study, which complied with all relevant regulations.' For more information on PLOS ONE's requirements for palaeontology and archaeology research, see https://journals.plos.org/plosone/s/submission-guidelines#loc-paleontology-and-archaeology-research

4. PLOS requires an ORCID iD for the corresponding author in Editorial Manager on papers submitted after December 6th, 2016. Please ensure that you have an ORCID iD and that it is validated in Editorial Manager. To do this, go to ‘Update my Information’ (in the upper left-hand corner of the main menu), and click on the Fetch/Validate link next to the ORCID field. This will take you to the ORCID site and allow you to create a new iD or authenticate a pre-existing iD in Editorial Manager. Please see the following video for instructions on linking an ORCID iD to your Editorial Manager account: https://www.youtube.com/watch?v=_xcclfuvtxQ.

“This research was supported by the National Natural Science Foundation of China (41890841, U1812402), the Project of The Department of Science and Technology of Guizhou Province (Guizhou Science and Technology Cooperation Platform Talents [2018]5613), the Study on metallogenic regularity and prospecting prediction of rare earth, barium, fluorine and other special resources in Guizhou ([2022]ZD004), and the Postgraduate Innovation Fund of Guizhou Province (Guizhou Education Cooperation YJSCXJH [2019]040). The author would like to express his thanks to these institutions.”

“This research was supported by the National Natural Science Foundation of China (41890841, U1812402), the Project of The Department of Science and Technology of Guizhou Province (Guizhou Science and Technology Cooperation Platform Talents [2018]5613), the Study on metallogenic regularity and prospecting prediction of rare earth, barium, fluorine and other special resources in Guizhou ([2022]ZD004), and the Postgraduate Innovation Fund of Guizhou Province (Guizhou Education Cooperation YJSCXJH [2019]040). The author would like to express his thanks to these institutions.”

6. Thank you for stating the following financial disclosure:

“This research was supported by the National Natural Science Foundation of China (41890841, U1812402), the Project of The Department of Science and Technology of Guizhou Province (Guizhou Science and Technology Cooperation Platform Talents [2018]5613), the Study on metallogenic regularity and prospecting prediction of rare earth, barium, fluorine and other special resources in Guizhou ([2022]ZD004), and the Postgraduate Innovation Fund of Guizhou Province (Guizhou Education Cooperation YJSCXJH [2019]040). The author would like to express his thanks to these institutions.”

7. In your Data Availability statement, you have not specified where the minimal data set underlying the results described in your manuscript can be found. PLOS defines a study's minimal data set as the underlying data used to reach the conclusions drawn in the manuscript and any additional data required to replicate the reported study findings in their entirety. All PLOS journals require that the minimal data set be made fully available. For more information about our data policy, please see http://journals.plos.org/plosone/s/data-availability.

8 . Please include captions for your Supporting Information files at the end of your manuscript, and update any in-text citations to match accordingly. Please see our Supporting Information guidelines for more information: http://journals.plos.org/plosone/s/supporting-information.

9. We note that Figures 1 and 2 in your submission contain [map/satellite] images which may be copyrighted. All PLOS content is published under the Creative Commons Attribution License (CC BY 4.0), which means that the manuscript, images, and Supporting Information files will be freely available online, and any third party is permitted to access, download, copy, distribute, and use these materials in any way, even commercially, with proper attribution. For these reasons, we cannot publish previously copyrighted maps or satellite images created using proprietary data, such as Google software (Google Maps, Street View, and Earth). For more information, see our copyright guidelines: http://journals.plos.org/plosone/s/licenses-and-copyright.

   a. You may seek permission from the original copyright holder of Figures 1 and 2 to publish the content specifically under the CC BY 4.0 license. 

Reviewers' comments:

Reviewer's Responses to Questions

**Comments to the Author**

1. Is the manuscript technically sound, and do the data support the conclusions?

Reviewer #1: Partly

Reviewer #2: Yes

2. Has the statistical analysis been performed appropriately and rigorously? 

Reviewer #1: No

Reviewer #2: Yes

3. Have the authors made all data underlying the findings in their manuscript fully available?

Reviewer #1: Yes

Reviewer #2: Yes

4. Is the manuscript presented in an intelligible fashion and written in standard English?

Reviewer #1: Yes

Reviewer #2: Yes

5. Review Comments to the Author

Reviewer #1: Dear Editor,

The ms “Studies on geochemical characteristics and biomineralization of Cambrian

phosphorites, Zhijin, Guizhou Province, China » deals with a comprehensive geochemical study on Cambrian phosphorites with more focus on the biomineralization expressed by the small shell fossils (SSFs). Although the topic is not new especially for the studied deposit where many studies were conducted but the discussion and modification appear to be worth of publishing. In my opinion, the major weakness point is that authors have not conducted geochemical analysis (major and Trace including REEs) on the small shell fossils (SSFs) but on the rock which is not really significant to decipher the link between hosting lithology and the SSFs. This led to a classic whole-rock geochemistry which SEM-EDS, being semi-quantitative analysis, could not be used as a significant data for the geochemical behavior of SSFs.

Another issues, is a mineralogical study is missing. In my opinion, mineralogy should conducted using X-ray or other techniques such as FTIR, if not micro-probe analysis (after formulas calculations) should be used rather than SEM-EDS is only semi-quantitative and can just used for overall (general) assessment. Additionally authors should carefully check unites as appended below. In my opinion the ms should be revised in a deep way before that can be accepted.

Specific comments:

Line 23-29 : long sentence, please reword.

Line 41 -42 : the same sentence is the abstract, could you please rephrase

Line 61: Authors mean “carbonate-fluor Apatite”? Please check and correct

Line 64: these only few studies among a large literature so please precede the cited references by (e.g.)

Line 81-84: add references

Line 89: you may change the term “wing” by “limb”, correct if necessary for a common terminology

Fig. 2: Extend acronyms in caption to

mining areas in Zhijin (Xiongjaichang (XJC), Lianxing (LX), Yinchanggou (YCG), Damachang

(DMC), Gaoshan (GS), and Gezhongwu (GZW) and define acronyms in the map like Z3, P2. ….

Line 105-108: Add reference for each sentence.

Fig. 3 (include acronym with the name of each profile to be well matched with previous maps). Additionally, figures are not clear, could you zoom the figure even below the profiles.

Line 137-139: Authors should add more details (material and methods) regarding trace and REE elements determination as well as accuracy, used standards and detection limits.

Line 203-205: Authors reported carbon “C” based on SEM-EDS. Authors should double check if the reported contents are not a result of C-metallization during preparing samples. Please indicate the sample preparation of SEM-EDS analysis in methods section.

Table 1. Correct “simple” to sample and check the unit of Al and better to replace “%” by “wt%” as a common unit.

Line 233: the reported units are wrong the content is in ppm why you add (10-6). Please correct all the mistakes in unites in all the ms.

Fig. 8 please indicate the used reference of the binary diagrams.

Line 247 -258: This could be more profitable in methods section. Please move it there.

Line 306-307. Add reference.

386-388: the section related to Ce anomaly should be improved furthermore the check the Ce reliability in light of new research such as https://doi.org/10.1016/j.gca.2022.03.003. Where some diagrams such as Y/Ho vs Ce/Ce* (correlations) should be tested. For Redox conditions and The grain size on REE uptake refer to the newest works such as: https://www.sciencedirect.com/science/article/abs/pii/S0375674222001169

- Correlations values: Authors should test the significance of correlations using p-values. Without a test, the correlations have no meaning (eg. Lines, 370, 393-394, and in the corresponding table.

- Line 520: Authors say that the main composition of the SSFs in the Zhijin phosphorites is carbonate-fluorapatite, this could be checked and used with caution as no mineralogical study are micro-probe analyses have conducted on separted SSFs?

Line 533: this a repetition, reword please.

The section 5.5 Metallogenic mechanism of the phosphorite block is missing references in many statements that authors should taking into account.

Line 599: Authors mention U/Th ratios although they do not discuss it in results or discussion?

Reviewer #2: RE: Studies on geochemical characteristics and biomineralization of Cambrian phosphorites, Zhijin, Guizhou Province, China

In this manuscript, the authors have investigated the geochemical features and biomineralization process of Cambrian Phosphorites in the Zhijin region of Guizhou Province, China. This manuscript is well organized and of high quality. I recommend this manuscript for acceptance and publication in the journal after minor revision:

1- In the sentences where several references are used in the manuscript text, it is mandatory to write the references in order from old to new.

2- It is better to write the words “Lower” and “Upper” as “lower” and “upper” in the whole manuscript. For example lower Cambrian.

3- Line 310: Please add the reference Abedini and Calagari (2017) to the end of the sentence and add it to the list of references.

Abedini A, Calagari A A, 2017. Geochemistry of claystones of the Ruteh Formation, NW Iran: Implications for provenance, source-area weathering, and paleo-redox conditions. Neues Jahrbuch für Mineralogie - Abhandlungen 194, 107–123.

4- Please use REE and REY throughout the manuscript. Avoid writing REEs and REYs.

5- Line 362, 348, 358, 369, 374....: Please write CaLagari as Calagari.

6- Line 465: Please add the reference Abedini et al. (2020) at the end of the sentence and add it to the list of references.

Abedini A, Rezaei Azizi M, Dill H G, 2020. Formation mechanisms of lanthanide tetrad effect in limestones: an example from Arbanos district, NW Iran. Carbonates and Evaporites 35 (1), 1-18. https://doi.org/10.1007/s13146-019-00533-z.

7- In general, the quality of the figures is low and the writing on the figures is illegible. It is better to prepare and paste them with high resolution.

8- Figure 11b should be deleted. In this form, the distribution of samples is very unusual.....

6. PLOS authors have the option to publish the peer review history of their article (what does this mean?). If published, this will include your full peer review and any attached files.

Reviewer #1: No

Reviewer #2: No

---

## [Author Response · Author response to Decision Letter 0]

13 Jan 2023

Editor’s comments:

Editor: Dear Dr. Yang,

Thank you for submitting your manuscript to PLOS ONE. I apologize for the long review process since I have had difficulty in finding appropriate reviewers for your work. After careful consideration, we feel that this manuscript is well-written with very good quality, but it does not fully meet PLOS ONE’s publication criteria as it currently stands. Therefore, we invite you to submit a revised version of the manuscript that addresses the points raised during the review process.

The reviewers suggest significant revisions on the introduction, methods, results and discussion parts of the manuscript, as well as modification of figures. I would recommend you considering the comments from the reviewers and provide a point-by-point response. Please note that the revised manuscript may be re-reviewed before considering for publication.

We look forward to receiving your revised manuscript.

Response: Thank you very much for your constructive comments and suggestions. They are valuable and helpful for improving our paper. In our revised manuscript, we have done our best to extensively revise the manuscript based on the constructive comments and suggestions from the editors and reviewers, and responded to all comments point by point as follows. All these changes have been highlighted in the Revised Manuscript with Track Changes.

Comment 1: Please ensure that your manuscript meets PLOS ONE's style requirements, including those for file naming. The PLOS ONE style templates can be found at

and

Response: Thanks for your helpful suggestion. In our resubmitted manuscript, the manuscript has been carefully checked to ensure compliance with the journal format. In addition, the Table 1, Table 2 and Table 3 from the original manuscript have been moved to Supporting information and uploaded as S1 Table, S2 Table and S3 Table. In our resubmitted manuscript, the locations of Table 1, Table 2 and Table 3 were replaced with S1 Table, S2 Table and S3 Table, respectively. 

Comment 2: We suggest you thoroughly copyedit your manuscript for language usage, spelling, and grammar. If you do not know anyone who can help you do this, you may wish to consider employing a professional scientific editing service.

Whilst you may use any professional scientific editing service of your choice, PLOS has partnered with both American Journal Experts (AJE) and Editage to provide discounted services to PLOS authors. Both organizations have experience helping authors meet PLOS guidelines and can provide language editing, translation, manuscript formatting, and figure formatting to ensure your manuscript meets our submission guidelines. To take advantage of our partnership with AJE, visit the AJE website (http://aje.com/go/plos) for a 15% discount off AJE services. To take advantage of our partnership with Editage, visit the Editage website (www.editage.com) and enter referral code PLOSEDIT for a 15% discount off Editage services. If the PLOS editorial team finds any language issues in text that either AJE or Editage has edited, the service provider will re-edit the text for free.

Response: Thanks for your helpful suggestion. Before submitting, the manuscript was edited in wording, punctuation, language, tense, and grammar by a highly qualified native English-speaking geoscientist. The certificate of language editing is shown as follows: 

Comment 3: In your Methods section, please provide additional information regarding the permits you obtained for the work. Please ensure you have included the full name of the authority that approved the field site access and, if no permits were required, a brief statement explaining why.

Additionally, in your manuscript, please provide additional information regarding the specimens used in your study. Ensure that you have reported specimen numbers and complete repository information, including museum name and geographic location. If permits were required, please ensure that you have provided details for all permits that were obtained, including the full name of the issuing authority, and add the following statement: 'All necessary permits were obtained for the described study, which complied with all relevant regulations.' If no permits were required, please include the following statement: 'No permits were required for the described study, which complied with all relevant regulations.' For more information on PLOS ONE's requirements for palaeontology and archaeology research, see https://journals.plos.org/plosone/s/submission-guidelines#loc-paleontology-and-archaeology-research

Response: Thanks for your helpful suggestion. In our resubmitted manuscript, according to your suggestion, the Permission Statement was added in Line 141~Line 142 in Section Material and methods, which is shown as follows:

Line 141~Line 142: “No permits were required for the described study, which complied with all relevant regulations.”

Comment 4: PLOS requires an ORCID iD for the corresponding author in Editorial Manager on papers submitted after December 6th, 2016. Please ensure that you have an ORCID iD and that it is validated in Editorial Manager. To do this, go to ‘Update my Information’ (in the upper left-hand corner of the main menu), and click on the Fetch/Validate link next to the ORCID field. This will take you to the ORCID site and allow you to create a new iD or authenticate a pre-existing iD in Editorial Manager. Please see the following video for instructions on linking an ORCID iD to your Editorial Manager account: https://www.youtube.com/watch?v=_xcclfuvtxQ.

Response: Thanks for your helpful suggestion. The ORCID iD of the corresponding author has been validated in Editorial Manager.

Comment 5: Thank you for stating the following in the Acknowledgments Section of your manuscript:

“This research was supported by the National Natural Science Foundation of China (41890841, U1812402), the Project of The Department of Science and Technology of Guizhou Province (Guizhou Science and Technology Cooperation Platform Talents [2018]5613), the Study on metallogenic regularity and prospecting prediction of rare earth, barium, fluorine and other special resources in Guizhou ([2022]ZD004), and the Postgraduate Innovation Fund of Guizhou Province (Guizhou Education Cooperation YJSCXJH [2019]040). The author would like to express his thanks to these institutions.”

“This research was supported by the National Natural Science Foundation of China (41890841, U1812402), the Project of The Department of Science and Technology of Guizhou Province (Guizhou Science and Technology Cooperation Platform Talents [2018]5613), the Study on metallogenic regularity and prospecting prediction of rare earth, barium, fluorine and other special resources in Guizhou ([2022]ZD004), and the Postgraduate Innovation Fund of Guizhou Province (Guizhou Education Cooperation YJSCXJH [2019]040). The author would like to express his thanks to these institutions.”

Response: Thanks for your helpful suggestion. We are very sorry for our insufficient consideration regarding the funding information. In our resubmitted manuscript, according to your suggestion, we have removed any funding-related text from the manuscript. In addition, the amended Funding Statement has been added in the cover letter, and please editor to update Funding Statement on our behalf. The amended Funding Statement is as follows:

This research was supported by the National Natural Science Foundation of China (41890841, U1812402), the Project of The Department of Science and Technology of Guizhou Province (Guizhou Science and Technology Cooperation Platform Talents [2018]5613), the Study on metallogenic regularity and prospecting prediction of rare earth, barium, fluorine and other special resources in Guizhou ([2022]ZD004), and the Postgraduate Innovation Fund of Guizhou Province (Guizhou Education Cooperation YJSCXJH [2019]040). The funders had no role in study design, data collection and analysis, decision to publish, or preparation of the manuscript.

Comment 6: Thank you for stating the following financial disclosure:

“This research was supported by the National Natural Science Foundation of China (41890841, U1812402), the Project of The Department of Science and Technology of Guizhou Province (Guizhou Science and Technology Cooperation Platform Talents [2018]5613), the Study on metallogenic regularity and prospecting prediction of rare earth, barium, fluorine and other special resources in Guizhou ([2022]ZD004), and the Postgraduate Innovation Fund of Guizhou Province (Guizhou Education Cooperation YJSCXJH [2019]040). The author would like to express his thanks to these institutions.”

Response: Thanks for your helpful suggestion. In our resubmitted manuscript, according to your suggestion, the role of the funder was added in the Funding Statement, and please editor to update Funding Statement on our behalf. The amended Funding Statement is as follows:

This research was supported by the National Natural Science Foundation of China (41890841, U1812402), the Project of The Department of Science and Technology of Guizhou Province (Guizhou Science and Technology Cooperation Platform Talents [2018]5613), the Study on metallogenic regularity and prospecting prediction of rare earth, barium, fluorine and other special resources in Guizhou ([2022]ZD004), and the Postgraduate Innovation Fund of Guizhou Province (Guizhou Education Cooperation YJSCXJH [2019]040). The funders had no role in study design, data collection and analysis, decision to publish, or preparation of the manuscript.

Comment 7: In your Data Availability statement, you have not specified where the minimal data set underlying the results described in your manuscript can be found. PLOS defines a study's minimal data set as the underlying data used to reach the conclusions drawn in the manuscript and any additional data required to replicate the reported study findings in their entirety. All PLOS journals require that the minimal data set be made fully available. For more information about our data policy, please see http://journals.plos.org/plosone/s/data-availability.

Response: Thanks for your helpful suggestion. We are very sorry for our insufficient consideration regarding the Data Availability Statement. According to your suggestion, our study’s minimal underlying data set have been uploaded as the Supporting Information files. In our resubmitted manuscript, the Section Supporting information was added in Line 708~Line 714, which is shown as follows:

Line 708~Line 714: Supporting information

S1 Table. Concentrations of major elements (wt%) of the Zhijin phosphorites samples in the lower Cambrian strata.

S2 Table. Concentrations of trace and rare earth elements (ppm) of the Zhijin phosphorites samples in the lower Cambrian strata.

S3 Table. Calculated values of some geochemical parameters of the Zhijin phosphorites samples in the lower Cambrian strata.

In addition, please editor to update the Data Availability Statement on our behalf. The Data Availability Statement is as follows:

Data Availability Statement: All relevant data are within the paper and its Supporting Information files.

Comment 8: Please include captions for your Supporting Information files at the end of your manuscript, and update any in-text citations to match accordingly. Please see our Supporting Information guidelines for more information: http://journals.plos.org/plosone/s/supporting-information.

Response: Thanks for your helpful suggestion. In our resubmitted manuscript, according to your suggestion, the Supporting information was added in the end of the manuscript in Line 708~Line 714, which is shown as follows:

Line 708~Line 714: Supporting information

S1 Table. Concentrations of major elements (wt%) of the Zhijin phosphorites samples in the lower Cambrian strata.

S2 Table. Concentrations of trace and rare earth elements (ppm) of the Zhijin phosphorites samples in the lower Cambrian strata.

S3 Table. Calculated values of some geochemical parameters of the Zhijin phosphorites samples in the lower Cambrian strata.

Comment 9: We note that Figures 1 and 2 in your submission contain [map/satellite] images which may be copyrighted. All PLOS content is published under the Creative Commons Attribution License (CC BY 4.0), which means that the manuscript, images, and Supporting Information files will be freely available online, and any third party is permitted to access, download, copy, distribute, and use these materials in any way, even commercially, with proper attribution. For these reasons, we cannot publish previously copyrighted maps or satellite images created using proprietary data, such as Google software (Google Maps, Street View, and Earth). For more information, see our copyright guidelines: http://journals.plos.org/plosone/s/licenses-and-copyright.

 a. You may seek permission from the original copyright holder of Figures 1 and 2 to publish the content specifically under the CC BY 4.0 license. 

Response: Thanks for your helpful suggestion. In our resubmitted manuscript, according to your suggestion, we have added references to Figures 1 and 2 in the figure titles of Line 89~Line 94, Line 110~Line 115 respectively, which is shown as follows:

Line 89~Line 94: “Fig 1. Map of China (a), Map of Guizhou Province (b), Regional geologic map of the Zhijin area showing the six sampling profiles, including the Xiongjaichang (XJC), Lianxing (LX), Yinchanggou (YCG), Damachang (DMC), Gaoshan (GS), and Gezhongwu (GZW) locations (c). (a) and (b) were created using free vector map data from the Public Domain (CC0), derived from USGS National Map Viewer (http://viewer.nationalmap.gov/viewer/). (c) modified after Fig 1 in ref. [3].”

Line 110~Line 115: “Fig 2. Lithofacies and paleogeography of the lower Cambrian strata in Zhijin showing: (a) Lithofacies and paleogeography of the lower Cambrian strata in South China Block; (b) Map of the paleogeographic zoning of the lower Cambrian phosphorite rocks in Zhijin area, including the Xiongjaichang (XJC), Lianxing (LX), Yinchanggou (YCG), Damachang (DMC), Gaoshan (GS), and Gezhongwu (GZW) sampling locations. (a) modified after Fig 1 in ref. [21]. (b) modified after Fig 3-9 in ref. [24].”

Reviewer#1 comments:

Reviewer #1: Dear Editor,

The ms “Studies on geochemical characteristics and biomineralization of Cambrian phosphorites, Zhijin, Guizhou Province, China » deals with a comprehensive geochemical study on Cambrian phosphorites with more focus on the biomineralization expressed by the small shell fossils (SSFs). Although the topic is not new especially for the studied deposit where many studies were conducted but the discussion and modification appear to be worth of publishing. In my opinion, the major weakness point is that authors have not conducted geochemical analysis (major and Trace including REEs) on the small shell fossils (SSFs) but on the rock which is not really significant to decipher the link between hosting lithology and the SSFs. This led to a classic whole-rock geochemistry which SEM-EDS, being semi-quantitative analysis, could not be used as a significant data for the geochemical behavior of SSFs.

Another issues, is a mineralogical study is missing. In my opinion, mineralogy should conducted using X-ray or other techniques such as FTIR, if not micro-probe analysis (after formulas calculations) should be used rather than SEM-EDS is only semi-quantitative and can just used for overall (general) assessment. Additionally authors should carefully check unites as appended below. In my opinion the ms should be revised in a deep way before that can be accepted.

Response: Thank you very much for your supportive and constructive comments. They are valuable for improving our paper and future research. According to these comments and suggestions, we have done our best to extensively revise the manuscript and provided further information point by point as follows. In addition, we checked the full text carefully and improved the expression of many words and sentences. All these changes have been highlighted in the Revised Manuscript with Track Changes.

As you said, the geochemical characteristics (including major, trace and rare earth elements) of small shell fossils (SSFs) is one of the important factors to explain the relationships between the organisms and the formation of phosphorites. However, the primary objectives of our study are to restore the phosphorus-formation environment of the Cambrian Meishucun Formation, and construct a sedimentary model of the phosphorites in the Meishucun Formation. Therefore, detailed geochemical investigations for small shell fossils (SSFs) will be our focus in future studies. In addition, the results of XRD analysis were carried out to examine the mineralogical composition of the SSFs, bioclastic phosphate rocks, siliceous phosphorite blocks and carbonaceous phosphorite blocks in the Zhijin area, which has been added to Line 250~Line 258 and shown as follows:

Line 258~Line 258: “The results of XRD analysis indicate that the SSFs are mostly composed of carbonate-fluorapatite, and small amount of quartz (Fig 6a). The bioclastic phosphate rocks in the DMC profile, LX profile and YCG profile are mainly composed of quartz and carbonate-fluorapatite, followed by illite, dolomite and jeremeievite (Figs 6b-d). In contrast, the siliceous phosphorite blocks of the GZW profile consist of dolomite, quartz and carbonate-fluorapatite (Fig 6e), and the carbonaceous phosphorite blocks of the LX profile consist of dolomite, quartz, carbonate-fluorapatite, pyrite and gypsum (Fig 6f). According to the XRD analysis, and combined with the SEM-EDS analysis, the mineralogical composition of the SSFs is dominated by carbonate-fluorapatite.”

Specific comments:

Comment 1-Line 23-29: long sentence, please reword.

Response: Thanks for your helpful suggestion. According to your suggestion, this sentence was rewritten in Line 22~Line 27 in our resubmitted manuscript, which is shown as follows:

Line 22~Line 27: “This study focuses on the biological fossil assemblage, the characteristics of phosphorus, and the relationship between biological and phosphorus enrichment of the lower Cambrian phosphorites. The primary objectives of our study are to analyze the role of organisms in the formation of phosphorites, restore the phosphorus-formation environment of the Cambrian Meishucun Formation, and construct a sedimentary model of the deposition of the phosphorites in the Meishucun Formation.”

Comment 2-Line 41 -42: the same sentence is the abstract, could you please rephrase.

Response: Thanks for your helpful suggestion. According to your suggestion, this sentence was rewritten in Line 36~Line 37 in our resubmitted manuscript, which is shown as follows:

Line 36~Line 37: “As an important ore resource in Guizhou Province, China, the phosphate rocks mainly occur in the Sinian Doushantuo Formation and the Cambrian Meishucun Formation.”

Comment 3-Line 61: Authors mean “carbonate-fluor Apatite”? Please check and correct.

Response: Thanks for your helpful suggestion. We are very sorry for our spelling mistake. In the resubmitted manuscript, the word “carbon-fluorapatite” has been replaced with “carbonate-fluorapatite” in Line 51, Line 243, Line 251~Line 252, Line 253, Line 255, Line 256, Line 258, Line 265, Line 604, Line 606, Line 617, and Line 618.

Comment 4-Line 64: these only few studies among a large literature so please precede the cited references by (e.g.)

Response: Thanks for your helpful suggestion. In our resubmitted manuscript, according to your suggestion, the word “e.g.” has been added to the front of the cited references in Line 54~Line 55, which is shown as follows:

Line 54~Line 55: “Geochemical methods are some of the most important methods in geological research. Previous studies (e.g., Emsbo et al., 2015; Fan et al., 2016; Chen et al., 2019) ……”

Comment 5-Line 81-84: add references

Response: Thanks for your helpful suggestion. In our resubmitted manuscript, according to your suggestion, the corresponding references have been added to the end of the sentence in Line 70~Line 73, which is shown as follows:

Line 70~Line 73: “The Zhijin phosphate deposit region was formed in the Meishucun period of the Cambrian (Xing et al., 2021). This deposit contains large-scale, high-quality phosphorus block rocks and represents the most important phosphorus formation period after the deposition of the phosphorus block rocks in China during the Doushantuo period (Chen et al., 2013; Zhou et al., 2019; Gong et al., 2021; Xing et al., 2021).”

References: 

Chen JY, Yang RD, Wei HR, Gao JB. Rare earth element geochemistry of Cambrian phosphorites from the Yangtze Region. Journal of Rare Earths. 2013; 31:101–112.

Gong XX, Wu SW, Xia Y, Zhang ZW, He S, Xie ZJ, et al. Enrichment characteristics and sources of the critical metal yttrium in Zhijin rare earth-containing phosphorites, Guizhou Province, China. Acta Geochimica. 2021; 40(3):25.

Xing JQ, Jiang YH, Xian HY, Zhang ZY, Yang YP, Tan W, et al. Hydrothermal activity during the formation of REY-rich phosphorites in the early Cambrian Gezhongwu Formation, Zhijin, South China: A micro- and nano-scale mineralogical study. Ore Geology Reviews. 2021; 136:104224.

Zhou KL, Fu Y, Ye YM, Long KS, Zhou WX. Characteristics of the Rare Earth Elements’ Accumulation of phosphorus rock series during the Early Cambrian, Guizhou Province. Acta Mineral Sinica. 2019; 39(4):420–431.

Comment 6-Line 89: you may change the term “wing” by “limb”, correct if necessary for a common terminology

Response: Thanks for your helpful suggestion. In our resubmitted manuscript, according to your suggestion, the term “wing” was replaced with “limb” in Line 77, which is shown as follows:

Line 77: “…and the research areas are mainly distributed in the northwestern limb of the Guohua and Zhangwei anticlines”

Comment 7-Fig. 2: Extend acronyms in caption to

mining areas in Zhijin (Xiongjaichang (XJC), Lianxing (LX), Yinchanggou (YCG), Damachang

(DMC), Gaoshan (GS), and Gezhongwu (GZW) and define acronyms in the map like Z3, P2. ….

Response: Thanks for your helpful suggestion. In our resubmitted manuscript, according to your suggestion, the full words of the abbreviations for the sampling locations in the figure 1 and figure2 has been added to the figure captions in Line 89~Line 94 and Line 110~Line 115, which is shown as follows:

Line 89~Line 94: “Map of China (a), Map of Guizhou Province (b), Regional geologic map of the Zhijin area showing the six sampling profiles, including the Xiongjaichang (XJC), Lianxing (LX), Yinchanggou (YCG), Damachang (DMC), Gaoshan (GS), and Gezhongwu (GZW) locations (c). (a) and (b) were created using free vector map data from the Public Domain (CC0), derived from USGS National Map Viewer (http://viewer.nationalmap.gov/viewer/). (c) modified after Fig 1 in ref. (Gao et al., 2018).”

Line 110~Line 115: “Fig 2. Lithofacies and paleogeography of the lower Cambrian strata in Zhijin showing: (a) Lithofacies and paleogeography of the lower Cambrian strata in South China Block; (b) Map of the paleogeographic zoning of the lower Cambrian phosphorite rocks in Zhijin area, including the Xiongjaichang (XJC), Lianxing (LX), Yinchanggou (YCG), Damachang (DMC), Gaoshan (GS), and Gezhongwu (GZW) sampling locations. (a) modified after Fig 1 in ref. (Chen et al., 2013). (b) modified after Fig 3-9 in ref. (Mao, 2015).”

References: 

Gao L, Yang RD, Gao JB, Chen JY, Chen J. Micro-structural characteristics and the relationship with forming phosphorus of the small shell fossils in Cambrian Meishucun stage phosphorites, XJC, Guizhou. Acta Palaeontologica Sinica. 2018; 57(3):273–286.

Chen JY, Yang RD, Wei HR, Gao JB. Rare earth element geochemistry of Cambrian phosphorites from the Yangtze Region. Journal of Rare Earths. 2013; 31:101–112.

Mao T. Analysis on the environment of phosphorus formation and controlling factors of mineralization in the bottom of Cambrian in central Guizhou. D.Sc. Thesis, Guizhou University, 2015.

Comment 8-Line 105-108: Add reference for each sentence.

Response: Thanks for your helpful suggestion. In our resubmitted manuscript, according to your suggestion, the corresponding references have been added to the end of the sentence in Line 100~Line 104, which is shown as follows:

Line 100~Line 104: “During the Meishucun period of the lower Cambrian in Zhijin, the depth of the sea water increased continuously during a transgression event (Chen et al., 2013; Xing et al., 2021). A large amount of phosphatic material was supplied by the stronger currents in the Meishucun period (Chen et al., 2013; Mao, 2015; Xing et al., 2021). The deposition process was controlled by many factors, including the biological chemistry and sedimentary environment (Pu et al., 1993; Chen et al., 2013; Zhou et al., 2019; Gong et al., 2021; Xing et al., 2021).”

References: 

Chen JY, Yang RD, Wei HR, Gao JB. Rare earth element geochemistry of Cambrian phosphorites from the Yangtze Region. Journal of Rare Earths. 2013; 31:101–112.

Gong XX, Wu SW, Xia Y, Zhang ZW, He S, Xie ZJ, et al. Enrichment characteristics and sources of the critical metal yttrium in Zhijin rare earth-containing phosphorites, Guizhou Province, China. Acta Geochimica. 2021; 40(3):25.

Mao T. Analysis on the environment of phosphorus formation and controlling factors of mineralization in the bottom of Cambrian in central Guizhou. D.Sc. Thesis, Guizhou University, 2015.

Xing JQ, Jiang YH, Xian HY, Zhang ZY, Yang YP, Tan W, et al. Hydrothermal activity during the formation of REY-rich phosphorites in the early Cambrian Gezhongwu Formation, Zhijin, South China: A micro- and nano-scale mineralogical study. Ore Geology Reviews. 2021; 136:104224.

Zhou KL, Fu Y, Ye YM, Long KS, Zhou WX. Characteristics of the Rare Earth Elements’ Accumulation of phosphorus rock series during the Early Cambrian, Guizhou Province. Acta Mineral Sinica. 2019; 39(4):420–431.

Comment 9-Fig. 3 (include acronym with the name of each profile to be well matched with previous maps). Additionally, figures are not clear, could you zoom the figure even below the profiles.

Response: Thanks for your helpful suggestion. In our resubmitted manuscript, according to your suggestions, the Fig 3 has been replaced with the following figure:

Comment 10-Line 137-139: Authors should add more details (material and methods) regarding trace and REE elements determination as well as accuracy, used standards and detection limits.

Response: Thanks for your helpful suggestion. We are very sorry for our insufficient consideration regarding the analytical method and detection limits of detecting trace and rare earth elements. In our resubmitted manuscript, the detailed analytical method and detection limits for trace and rare earth elements have been provided in Line 167~Line 174, which is shown as follows:

Line 167~Line 174: “The trace and rare earth elements of 41 samples were analyzed by using Quadrupole inductively coupled plasma-mass spectrometer (Q-ICP-MS, ELAN DRC-e, PerkinElmer, Canada). The analysis method was follows: the powdered samples (~200 mesh) were digested with HCl, HNO3, HF and HClO4, after dryness, prepared samples were dissolved with HCl, and then analyzed by Q-ICP-MS. The detection limits of trace and rare earth elements were: for V, Ni, Ag, As and Sb 0.01 ppm; For Y and REE 0.01 to 0.05 ppm. The trace and rare earth elements of samples were analyzed at the Institute of Geochemistry, Chinese Academy of Sciences.”

Comment 11-Line 203-205: Authors reported carbon “C” based on SEM-EDS. Authors should double check if the reported contents are not a result of C-metallization during preparing samples. Please indicate the sample preparation of SEM-EDS analysis in methods section.

Response: Thanks for your helpful suggestion. In our resubmitted manuscript, according to your suggestions, the description of sample preparation of SEM-EDS analysis was added to Line 154~Line 155, which is shown as follows:

Line 154~Line 155: “Before the testing of SEM-EDS, the fossil shells and thin sections were sprayed with a thin layer of gold.”

Comment 12-Table 1. Correct “simple” to sample and check the unit of Al and better to replace “%” by “wt%” as a common unit.

Response: Thanks for your helpful suggestion. We are very sorry for our spelling mistake. In our resubmitted manuscript, the “simple” in the S1 Table was replaced with “sample”. In addition, the unit of major elements was illustrated in table captions in Line 709~Line 710, which is shown as follows:

Line 709~Line 710: “S1 Table. Concentrations of major elements (wt%) of the Zhijin phosphorites samples in the lower Cambrian strata.”

Comment 13-Line 233: the reported units are wrong the content is in ppm why you add (10-6). Please correct all the mistakes in unites in all the ms.

Response: Thanks for your helpful suggestion. We are very sorry for our negligence. In our resubmitted manuscript, the units of trace and rare earth elements throughout the manuscript was unified into ppm.

Comment 14-Fig. 8 please indicate the used reference of the binary diagrams.

Response: Thanks for your helpful suggestion. In our resubmitted manuscript, according to your suggestion, the cited reference was added to the Figure caption in Line 418~Line 419, which is shown as follows:

Line 418~Line 419: “Fig 10. The binary diagrams of (a) V/Cr vs Mo and (b) Ni/Co vs Mo showing the redox conditions of the sedimentary seawater (after Jones and Manning, 1994).”

References: 

Jones B, Manning DA. Comparison of geochemical indices used for the interpretation of palaeoredox conditions in ancient mudstones. Chemical Geology. 1994; 111:111–129.

Comment 15-Line 247 -258: This could be more profitable in methods section. Please move it there.

Response: Thanks for your helpful suggestion. In our resubmitted manuscript, according to your suggestion, the calculation of REE parameters has been adjusted to the 3.1 Geochemical analysis section in Line 174~Line 178, which is shown as follows:

Line 174~Line 178: “The results of whole-rock major, trace and rare earth elements concentrations in all samples are reported in S1-S3 Tables. The REE parameters analyzed are as follows: δCe=Ce/Ce*=CeN/0.5(LaN+PrN) (Bau and Dulski, 1996), δEu=Eu/Eu*=EuN/(SmN×GdN)0.5 (Taylor and McLennan, 1985), Pr/Pr*=PrN/0.5(CeN+NdN) (Bau and Dulski, 1996), and Y/Y*=2YN/(DyN+HoN) (Fazio et al., 2007), where N denotes normalization relative to Post-Archean Australian Shale (PAAS) (Taylor and McLennan,1985).”

References: 

Bau M. Controls on fractionation of isovalent trace elements in magmatic and aqueous systems: evidence from Y/Ho, Zr/Hf and lanthanide tetrad effect. Contributions to Mineralogy & Petrology. 1996; 123:323–333.

Fazio AM, Scasso RA, Castro LN, Carey S. Geochemistry of rare earth elements in early-diagenetic Miocene phosphatic concretions of Patagonia, Argentina: phosphogenetic implications. Deep Sea Research Part II Topical Studies in Oceanography. 2007; 54:1414–1432.

Taylor SR, McLennan SM. The Continental Crust: its Composition and Evolution. Blackwell, 1985. P. 1–312.

Comment 16-Line 306-307. Add reference.

Response: Thanks for your helpful suggestion. In our resubmitted manuscript, according to your suggestion, the cited reference was added to Line 392~Line 393, which is shown as follows:

Line 392~Line 393: “V and Cr are soluble in water under oxic conditions, and they tend to be enriched in sediments deposited under reducing environment (Piper, 1994).”

References: 

Piper DZ, 1994. Seawater as the source of minor elements in black shales, phosphorites and other sedimentary rocks. Chemical Geology. 1994; 114(1–2):95–114.

Comment 17-386-388: the section related to Ce anomaly should be improved furthermore the check the Ce reliability in light of new research such as https://doi.org/10.1016/j.gca.2022.03.003. Where some diagrams such as Y/Ho vs Ce/Ce* (correlations) should be tested. For Redox conditions and The grain size on REE uptake refer to the newest works such as: https://www.sciencedirect.com/science/article/abs/pii/S0375674222001169

Response: Thanks for your helpful suggestion. In our resubmitted manuscript, according to the latest research (Zhang et al., 2022), the section related to Ce anomaly has been rewritten in Line 370~Line 371, Line 379~Line 386, which is shown as follows:

Line 370~Line 371: “It was found that the Ce anomalies of the Zhijin phosphorites plot in the IIIb domain, which belongs to the real Ce anomaly range (Fig 9a), it appears to reflect the phosphorites were formed under oxidic conditions.”

Line 379~Line 386: “However, many studies believe that the Ce anomalies in francolite may also involve REY fractionation during diagenetic uptake and relate to porewater REY chemistry, which cannot effectively constrain the redox environment (Herwartz et al., 2011; Liao et al., 2019; Zhang et al., 2022). If the poor correlation between Ce/Ce* values and Y/Ho ratios in phosphorites indicates that this principle is also applicable to phosphorites (Zhang et al., 2022). There is a poor correlation between Ce/Ce* values and Y/Ho ratio (R2=0.06, p>0.05) (Fig 9b), indicating that the Ce anomaly of Zhijin phosphorites may not only be affected by the REDOX state of seawater, but also may be caused by REY fractionation during diagenetic absorption. Therefore, the Ce anomaly cannot be completely used to limit the REDOX history of Zhijin phosphorites.”

In addition, the related discussion regarding the redox conditions and grain size on REE uptake was added to Line 339~Line 360 and shown as follows:

Line 339~Line 360: “Meanwhile, the distribution of REY in the phosphorite is mainly controlled by the sedimentary environment, but irrespective of grain size in the rock (Ferhaoui et al., 2022).”

References: 

Ferhaoui S, Kechiched R, Bruguier O, Sinisi R, Kocsis L, Mongelli G, et al. Rare earth elements plus yttrium (REY) in phosphorites from the Tébessa region (Eastern Algeria): Abundance, geochemical distribution through grain size fractions, and economic significance. Journal of Geochemical Exploration. 2022; 241:107058.

Herwartz D, Tu¨tken T, Mu¨nker C, Jochum KP, Stoll B, Sander PM. Timescales and mechanisms of REE and Hf uptake in fossil bones. Geochimica et Cosmochimica Acta. 2011; 75(1):82–105.

Liao JL, Sun XM, Li DF, Sa RN, Lu Y, Lin ZY, et al. New insights into nanostructure and geochemistry of bioapatite in REE-rich deep-sea sediments: LA-ICP-MS, TEM, and Z-contrast imaging studies. Chemical Geology. 2019; 512:58–68.

Zhang HJ, Fan HF, Wen HJ, Han T, Zhou T, Xia Y. Controls of REY enrichment in the early Cambrian phosphorites. Geochimica et Cosmochimica Acta. 2022; 324:117–139.

Comment 18- Correlations values: Authors should test the significance of correlations using p-values. Without a test, the correlations have no meaning (eg. Lines, 370, 393-394, and in the corresponding table.

Response: Thanks for your helpful suggestion. We are very sorry for our insufficient consideration regarding the test of correlations using p-values. In our resubmitted manuscript, all correlation analyses have been tested using p-values.

Comment 19- Line 520: Authors say that the main composition of the SSFs in the Zhijin phosphorites is carbonate-fluorapatite, this could be checked and used with caution as no mineralogical study are micro-probe analyses have conducted on separted SSFs?

Response: Thanks for your helpful suggestion. In our resubmitted manuscript, according to your suggestion, the XRD analysis was performed to determine the mineralogical composition of the SSFs, bioclastic phosphate rocks, siliceous phosphorite blocks and carbonaceous phosphorite blocks in the Zhijin area. In our resubmitted manuscript, the results of XRD analysis have been added to Line 251~Line 258, which is shown as follows:

Line 251~Line 258: “The results of XRD analysis indicate that the SSFs are mostly composed of carbonate-fluorapatite, and small amount of quartz (Fig 6a). The bioclastic phosphate rocks in the DMC profile, LX profile and YCG profile are mainly composed of quartz and carbonate-fluorapatite, followed by illite, dolomite and jeremeievite (Figs 6b-d). In contrast, the siliceous phosphorite blocks of the GZW profile consist of dolomite, quartz and carbonate-fluorapatite (Fig 6e), and the carbonaceous phosphorite blocks of the LX profile consist of dolomite, quartz, carbonate-fluorapatite, pyrite and gypsum (Fig 6f). According to the XRD analysis, and combined with the SEM-EDS analysis, the mineralogical composition of the SSFs is dominated by carbonate-fluorapatite.”

Comment 20- Line 533: this a repetition, reword please.

Response: Thanks for your helpful suggestion. In our resubmitted manuscript, according to your suggestion, this sentence was rewritten in Line 616~Line 617, which is shown as follows:

Line 616~Line 617: “In general, carbonate-fluorapatite is the dominant composition of the SSFs in the Zhijin phosphorites…”

Comment 21-The section 5.5 Metallogenic mechanism of the phosphorite block is missing references in many statements that authors should taking into account.

Response: Thanks for your helpful suggestion. In our resubmitted manuscript, according to your suggestion, the cited reference was added to Line 622~Line 681, which is shown as follows:

Line 622~Line 681: “Large-scale marine sedimentary phosphate deposits were formed in the Late Sinian Doushantuo stage and the Early Cambrian Meishucun stage on the Yangtze platform during two important stages of phosphorus formation (Chen et al., 2013, 2022; Mao, 2015; Zhou et al., 2019; Gong et al., 2021; Zhang et al., 2022). The phosphorus-bearing rock series in Zhijin was formed during the Meishucun period in the lower Cambrian (Xing et al., 2021; Zhang et al., 2022), and the formation of these phosphate deposits mainly included three factors: phosphorus source, migration, and pre-precipitation concentration (Mao, 2015; Gong et al., 2021; Liu et al., 2020; Yang et al., 2021). Based on the results obtained in this study and previous research data, the formation of the Zhijin phosphorous block has been summarized as follows (Fig 19).

Process A. Platform margin: The rising ocean currents transported abundant phosphorus and REE containing materials from the deep sea due to the large-scale transgression during the Meishucun period, and these currents reached the platform margin (Chen et al., 2013, 2022; Mao, 2015; Xing et al., 2021). Silt and other impurities in the high-phosphorus seawater were deposited first, resulting in a rapid increase in the P and Ca contents of the seawater. The temperature of the sea water increased accordingly, and the water also changed from a reducing environment to an oxidizing environment, so the organisms flourished in large quantities (Liu et al., 2020). Additionally, the biochemical activities lead to the precipitation of phosphate. At this time, the water at the edge of the platform was deep, which was not conducive to the deposition and mineralization of phosphorus, so phosphate deposits seldom formed during this process (Fig 20a) (Liu et al., 2020; Yang et al., 2021).

Process B. Shallow sedimentary environment: After process A, the transgression continued and the seawater reached the shoal area, which flooded the platform, making the water shallower, which was conducive to the deposition of phosphorus in the seawater (Liu et al., 2020). During process B, the O2 content increased, the sunlight was sufficient, and the marine organisms flourished (Zhang et al., 2006). Eventually, under the joint influence of the biological and chemical actions, the phosphate in the seawater reached the state of supersaturation, and phosphate deposits formed (Fig 20b).

Process C. Tidal flat deposits: The seawater reached the tidal flat sedimentary zone of the shallow sea and formed an oxidizing water environment. This was most suitable for the deposition of the phosphorite block, and the Zhijin phosphate deposit was formed in this stage (Fig 20) (Liu et al., 2020; Chen et al., 2022; Zhang et al., 2022). The phosphorus-rich water supplied by the upwelling current reached the shallow sea area where benthic algae mainly developed (Liu et al., 2020). As the phosphorus content of the water increased, owing to the suitable temperature, hydrodynamics, and O2 content of the sea water, small shell fauna prospered and absorbed phosphorus to form their shells (Zhu et al., 1996; Mi et al., 2011; Moysiuk et al., 2017). In addition, the adsorption of organisms also promoted the decomposition and precipitation of phosphorus from the phosphorus-containing organic matter (Fig 20c1) (Yang et al., 2021). With the flourishing biological activity, the physicochemical properties of the shallow sea area changed, becoming rich in a large number of phosphorus accumulating biogenic organisms, which were buried after death (Zhu et al., 1996; Xue et al., 2006; Gao et al., 2018). In addition, phosphorus adsorption by shells continued to play an important role, and the water soluble phosphorus enhancement promoted the accumulation of phosphatic material via chemical precipitation (Fig 20c2) (Zhang et al., 2006; Yang et al., 2021). Phosphorus was enriched and mineralized, and collophanite mineralization occurred in shallow sea water, forming a large-scale, high P2O5 grade phosphate deposit (Fig 20e) (Liu et al., 2020; Yang et al., 2021; Chen et al., 2022; Zhang et al., 2022).

Process D. Upper seawater environment: The phosphorus in the seawater was absorbed by the abundant plankton, and their death transported part of the phosphorus into the deep water (Fig 20d1) (Mao, 2015; Yang et al., 2021). The CO2 content of the deep seawater increased, facilitating the dissolution of phosphorus in the deep seawater (Fig 20d2) (Zhang et al., 2006). The phosphorus content of the water increased rapidly, and the seawater biomass continued to flourish. This process continued throughout the transgression phase (Chen, 2009; Liu et al., 2017).

References: 

Chen JY, Yang RD, Wei HR, Gao JB. Rare earth element geochemistry of Cambrian phosphorites from the Yangtze Region. Journal of Rare Earths. 2013; 31:101–112.

Chen JY, Yang RD, Zhang J, Chao JX. Occurrence of yttrium in the Zhijin phosphorus deposit in Guizhou Province, China. Open Geosciences. 2022; 14(1):776–784.

Chen M. Chemical oceanography. China Ocean Press; 2009. P. 1–265.

Gao L, Yang RD, Gao JB, Chen JY, Chen J. Micro-structural characteristics and the relationship with forming phosphorus of the small shell fossils in Cambrian Meishucun stage phosphorites, XJC, Guizhou. Acta Palaeontologica Sinica. 2018; 57(3):273–286.

Gong XX, Wu SW, Xia Y, Zhang ZW, He S, Xie ZJ, et al. Enrichment characteristics and sources of the critical metal yttrium in Zhijin rare earth-containing phosphorites, Guizhou Province, China. Acta Geochimica. 2021; 40(3):25.

Liu XQ, Zhang H, Tang Y, Liu YL. REE Geochemical Characteristic of Apatite: Implications for Ore Genesis of the Zhijin Phosphorite. Minerals. 2020; 10(11):1012.

Liu Z, Zhou M. Meishucun phosphorite succession (SW China) records redox changes of the early Cambrian ocean. Geological Society of America Bulletin. 2017; 129:1554–1567.

Mao T. Analysis on the environment of phosphorus formation and controlling factors of mineralization in the bottom of Cambrian in central Guizhou. D.Sc. Thesis, Guizhou University, 2015.

Mi WT, Li DL, Fan Y. Geochemical characteristics of the Doushantuo Formation phosphorite in Baiguoyuan Formation, Yichang, Hubei Province. Geology and Exploration. 2011; 47:982–993.

Moysiuk J, Smith MR, Caron JB. Hyoliths are Palaeozoic lophophorates. Nature. 2017; 541:394–398.

Xing JQ, Jiang YH, Xian HY, Zhang ZY, Yang YP, Tan W, et al. Hydrothermal activity during the formation of REY-rich phosphorites in the early Cambrian Gezhongwu Formation, Zhijin, South China: A micro- and nano-scale mineralogical study. Ore Geology Reviews. 2021; 136:104224.

Xue YS, Zhou CM. Re-sedimentation and stratigraphic correlation of early Cambrian phosphorous small shell fossils in the Yangtze region. Journal of Stratigraphy. 2006; 30(1):64–74.

Yang HY, Xiao JF, Xia Y, Xie ZJ, Tan QP, Xu JB, et al. Phosphorite generative processes around the Precambrian-Cambrian boundary in South China: An integrated study of Mo and phosphate O isotopic compositions. Geoscience Frontiers. 2021; 12(5):101187.

Zhang HJ, Fan HF, Wen HJ, Han T, Zhou T, Xia Y. Controls of REY enrichment in the early Cambrian phosphorites. Geochimica et Cosmochimica Acta. 2022; 324:117–139.

Zhang J, Zhu L, Zhang Q. Biological ore characteristic of ore-bearing REE in Xinhua phosphorite, Zhijin, Guizhou. Chinese Rare Earths. 2006; 27:93–94.

Zhou KL, Fu Y, Ye YM, Long KS, Zhou WX. Characteristics of the Rare Earth Elements’ Accumulation of phosphorus rock series during the Early Cambrian, Guizhou Province. Acta Mineral Sinica. 2019; 39(4):420–431.

Zhu MY, Qian Y, Jiang ZW, He TG. Primary discussion on preserving, composition and microstructure of small shelly fossil. Acta Micropalaeontologica Sinica. 1996; 13:241–254.

Comment 22-Line 599: Authors mention U/Th ratios although they do not discuss it in results or discussion?

Response: Thank you for pointing out this mistake. We are very sorry for our negligence. In our resubmitted manuscript, this conclusion was rewritten in Line 697~Line 701, which is shown as follows:

Line 697~Line 701: “(3) The REE content of the Zhijin biophosphate is very high, and this REE enrichment may be related to the activities of organisms. The calculation results of Ce anomaly, V/Cr ratio and Ni/Co ratio demonstrate that the lower Cambrian Zhijin phosphorites were mainly formed in oxidic environment. This oxygen-rich environment is conducive to the participation of organisms and to their subsequent mineralization.”

Reviewer#2 comments:

Reviewer #2: RE: Studies on geochemical characteristics and biomineralization of Cambrian phosphorites, Zhijin, Guizhou Province, China

In this manuscript, the authors have investigated the geochemical features and biomineralization process of Cambrian Phosphorites in the Zhijin region of Guizhou Province, China. This manuscript is well organized and of high quality. I recommend this manuscript for acceptance and publication in the journal after minor revision:

Response: Thank you very much for your supportive and constructive comments. They are valuable and helpful for improving our paper. According to these comments and suggestions, we have done our best to extensively revise the manuscript and provided further information point by point as follows. All these changes have been marked in the Revised Manuscript with Track Changes.

Comment 1- In the sentences where several references are used in the manuscript text, it is mandatory to write the references in order from old to new.

Response: Thank you very much for reminding us of this important point. In our resubmitted manuscript, we have carefully checked the manuscript and made sure that all the references are in order from old to new.

Comment 2- It is better to write the words “Lower” and “Upper” as “lower” and “upper” in the whole manuscript. For example lower Cambrian.

Response: Thanks for your constructive suggestion. According to your suggestion, in the resubmitted manuscript, the words “Lower” and “Upper” have been respectively replaced with “lower” and “upper” in our resubmitted manuscript.

Comment 3- Line 310: Please add the reference Abedini and Calagari (2017) to the end of the sentence and add it to the list of references.

Abedini A, Calagari A A, 2017. Geochemistry of claystones of the Ruteh Formation, NW Iran: Implications for provenance, source-area weathering, and paleo-redox conditions. Neues Jahrbuch für Mineralogie - Abhandlungen 194, 107–123.

Response: Thanks for your helpful suggestion. As you said, the reference Abedini and Calagari (2017) is very important to support the V/Cr ratio used in this manuscript. Therefore, in the resubmitted manuscript, the reference Abedini and Calagari (2017) has been respectively added to Line 396and the reference list.

References: 

Abedini A, Calagari AA. Geochemistry of claystones of the Ruteh Formation, NW Iran: Implications for provenance, source-area weathering, and paleo-redox conditions. Neues Jahrbuch für Mineralogie – Abhandlungen. 2017; 194:107–123.

Comment 4- Please use REE and REY throughout the manuscript. Avoid writing REEs and REYs.

Response:Thanks for your helpful suggestion. According to your suggestion, in the resubmitted manuscript, the abbreviations “REEs” and “REYs” have been respectively replaced with “REE” and “REY”.

Comment 5- Line 362, 348, 358, 369, 374....: Please write CaLagari as Calagari.

Response:Thanks for your helpful suggestion. We are very sorry for our spelling mistake. In the resubmitted manuscript, the name “CaLagari” has been replaced with “Calagari” in our resubmitted manuscript.

Comment 6- Line 465: Please add the reference Abedini et al. (2020) at the end of the sentence and add it to the list of references.

Abedini A, Rezaei Azizi M, Dill H G, 2020. Formation mechanisms of lanthanide tetrad effect in limestones: an example from Arbanos district, NW Iran. Carbonates and Evaporites 35 (1), 1-18. https://doi.org/10.1007/s13146-019-00533-z.

Response: Thanks for your helpful suggestion. As you said, the reference Abedini et al. (2020) is very important to support the REE distribution patterns as an indicator for evaluating the sedimentary environment. Therefore, in the resubmitted manuscript, the reference Abedini et al. (2020) has been respectively added to Line 550~Line 551 and the reference list.

References: 

Abedini A, Azizi MR, Dill HG. Formation mechanisms of lanthanide tetrad effect in limestones: an example from Arbanos district, NW Iran. Carbonates and Evaporites. 2020; 35(1):1–18.

Comment 7- In general, the quality of the figures is low and the writing on the figures is illegible. It is better to prepare and paste them with high resolution.

Response: Thank you very much for reminding us of this important point. As you said, most of the words in the figures of the original manuscript are too small to see. Therefore, in our resubmitted manuscript, the words and quality of the figures have been modified to ensure that they can be seen clearly. We would continue to revise it if the new version doesn’t meet the requirement.

Comment 8- Figure 11b should be deleted. In this form, the distribution of samples is very unusual.....

Response: Thanks for your helpful suggestion. As you said, the Figure 11b in the original manuscript is redundant and unusual in the Section 5.2.3 Diagenetic weathering and post-diagenesis reworking. Therefore, this figure has been deleted in the resubmitted manuscript.

---

## [Decision Letter · Decision Letter 1]

30 Jan 2023

Studies on geochemical characteristics and biomineralization of Cambrian phosphorites, Zhijin, Guizhou Province, China

PONE-D-22-21235R1

Dear Dr. Yang,

Thank you for submitting your revised manuscript with a detailed reply to address the reviewers' comments. After careful review, we’re pleased to inform you that your manuscript has been judged scientifically suitable for publication and will be formally accepted for publication once it meets all outstanding technical requirements.

Kind regards,

Ziming Yang, PhD

Academic Editor

PLOS ONE

Additional Editor Comments (optional):

Reviewers' comments:

Reviewer's Responses to Questions

**Comments to the Author**

1. If the authors have adequately addressed your comments raised in a previous round of review and you feel that this manuscript is now acceptable for publication, you may indicate that here to bypass the “Comments to the Author” section, enter your conflict of interest statement in the “Confidential to Editor” section, and submit your "Accept" recommendation.

Reviewer #1: All comments have been addressed

Reviewer #2: All comments have been addressed

2. Is the manuscript technically sound, and do the data support the conclusions?

Reviewer #1: Yes

Reviewer #2: Yes

3. Has the statistical analysis been performed appropriately and rigorously? 

Reviewer #1: Yes

Reviewer #2: Yes

4. Have the authors made all data underlying the findings in their manuscript fully available?

Reviewer #1: Yes

Reviewer #2: Yes

5. Is the manuscript presented in an intelligible fashion and written in standard English?

Reviewer #1: Yes

Reviewer #2: Yes

6. Review Comments to the Author

Reviewer #1: Dear Editor

Thank you for sending this new version. Authors have taken into account all my comments.

Reviewer #2: I carefully examined the manuscript (including text, tables and figures). The authors have greatly improved the manuscript and responded satisfactorily to all comments. Now the manuscript has been updated very well and professionally.Now the manuscript can be published in the journal.

7. PLOS authors have the option to publish the peer review history of their article (what does this mean?). If published, this will include your full peer review and any attached files.

Reviewer #1: No

Reviewer #2: No

---

## [Editor Report · Acceptance letter]

2 Feb 2023

PONE-D-22-21235R1 

Studies on geochemical characteristics and biomineralization of Cambrian phosphorites, Zhijin, Guizhou Province, China 

Dear Dr. Yang:

I'm pleased to inform you that your manuscript has been deemed suitable for publication in PLOS ONE. Congratulations! Your manuscript is now with our production department. 

Kind regards, 

on behalf of

Dr. Ziming Yang 

Academic Editor

PLOS ONE